# BP($\lambda$): Online Learning via Synthetic Gradients

**Joseph Pemberton**                                                    *joe.pemberton@bristol.ac.uk*
*Computational Neuroscience Unit, Faculty of Engineering, University of Bristol, United Kingdom*
*Centre for Neural Circuits and Behaviour, Department of Physiology, Anatomy and Genetics, Medical Sciences Division, University of Oxford, United Kingdom*

**Rui Ponte Costa**                                                    *rui.costa@dpag.ox.ac.uk*
*Centre for Neural Circuits and Behaviour, Department of Physiology, Anatomy and Genetics, Medical Sciences Division, University of Oxford, United Kingdom*
*Computational Neuroscience Unit, Faculty of Engineering, University of Bristol, United Kingdom*

**Reviewed on OpenReview:** *https://openreview.net/forum?id=3kYgouAfqk*

## Abstract

Training recurrent neural networks typically relies on backpropagation through time (BPTT). BPTT depends on forward and backward passes to be completed, rendering the network locked to these computations before loss gradients are available. Recently, Jaderberg et al. proposed synthetic gradients to alleviate the need for full BPTT. In their implementation synthetic gradients are learned through a mixture of backpropagated gradients and bootstrapped synthetic gradients, analogous to the temporal difference (TD) algorithm in Reinforcement Learning (RL). However, as in TD learning, heavy use of bootstrapping can result in bias which leads to poor synthetic gradient estimates. Inspired by the accumulate TD($\lambda$) in RL, we propose a fully online method for learning synthetic gradients which avoids the use of BPTT altogether: *accumulate BP($\lambda$)*. As in accumulate TD($\lambda$), we show analytically that accumulate BP($\lambda$) can control the level of bias by using a mixture of temporal difference errors and recursively defined eligibility traces. We next demonstrate empirically that our model outperforms the original implementation for learning synthetic gradients in a variety of tasks, and is particularly suited for capturing longer timescales. Finally, building on recent work we reflect on accumulate BP($\lambda$) as a principle for learning in biological circuits. In summary, inspired by RL principles we introduce an algorithm capable of bias-free online learning via synthetic gradients.

## 1 Introduction

A common approach for solving temporal tasks is to use recurrent neural networks (RNNs), which with the right parameters can effectively integrate and maintain information over time. The temporal distance between inputs and subsequent task loss, however, can make optimising these parameters challenging. The backpropagation through time (BPTT) algorithm is the classical solution to this problem that is applied once the task is complete and all task losses are propagated backwards in time through the preceding chain of computation. Exact loss gradients are thereby derived and are used to guide updates to the network parameters.

However, BPTT can be undesirably expensive to perform, with its memory and computational requirements scaling intractably with the task duration. Moreover, the gradients can only be obtained after the RNN forward and backward passes have been completed. This makes the network parameters effectively locked until those computations are carried out. One common solution to alleviate these issues is to apply truncated BPTT, where error gradients are only backpropagated within fixed truncation windows, but this approach can limit the network's ability to capture long-range temporal dependencies.

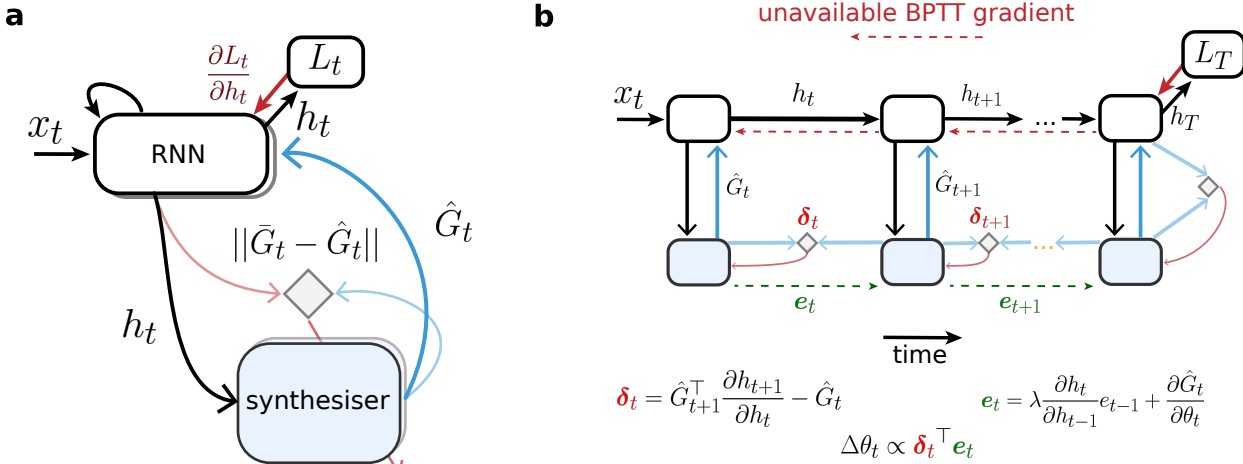

Figure 1: **Schematic of a recurrent neural network (RNN) which learns via synthetic gradients**. (**a**) External input $x_t$ is provided to the RNN which has hidden state $h_t$. Due to recurrency this state will affect the task loss at the current timestep $L_t$ and future timesteps $L_{>t}$ not yet seen. A distinct synthesiser network receives $h_t$ as input and estimates its future loss gradient $\hat{G}_t \approx \frac{\partial L_{>t}}{\partial h_t}$, which is provided to the RNN for learning. The synthesiser learns to mimic a target gradient $\bar{G}_t$. How $\bar{G}_t$ is defined and learned is the focus of this paper. (**b**) An illustration of the accumulate BP($\lambda$) algorithm for learning synthetic gradients in an unrolled version of the network. Current activity $h_t$ must be correctly associated to the later task loss $L_T$. Here the parameters $\theta$ of the synthesiser are updated via a mixture of temporal difference errors $\delta$ (red) and eligibility traces $e$ (green). As in accumulate TD($\lambda$) in RL, $\delta$ is computed online using bootstrapping whilst $e$ propagates forwards with a decay component $\lambda$ with $0 \leq \lambda \leq 1$. Together, they approximate the true loss gradient. In contrast to the original synthetic gradient algorithm by Jaderberg et al. (2017), our model does not require BPTT.

One proposed method which avoids the need for many-step BPTT whilst still capturing long-range dependencies is to apply synthetic gradients (Jaderberg et al., 2017; Czarnecki et al., 2017). In this method gradients from future errors are predicted by a separate network, a "synthesiser", given the current RNN activity (Fig. 1**a**). Synthetic gradients enable the network to model long-range dependencies on future errors whilst avoiding the waiting time imposed by BPTT. The independence of memory and computational complexity with respect to the total task length makes synthetic gradients an attractive alternative compared to BPTT (Marschall et al., 2020). Recently, these properties have led neuroscientists to speculate that synthetic gradients are computed at the systems-level in the brain, explaining a range of experimental observations (Marschall et al., 2019; Pemberton et al., 2021; Boven et al., 2023).

Despite their promise, the full potential of approximating BPTT with synthetic gradients has not yet been realised. In particular, it is not yet clear what are the optimal conditions for learning synthetic gradients. In its original implementation, Jaderberg et al. use synthetic gradients alongside truncated BPTT, and define the synthesiser target as a mixture of backpropagated gradients with its own predicted (future) gradient. That is, the synthesiser uses its own estimations – *bootstrapping* – for learning. As the original authors note, this is highly reminiscent of temporal difference (TD) algorithms used in Reinforcement Learning (RL) which use bootstrapping for estimating the future return (Sutton & Barto, 2018). Indeed, in their supplementary material Jaderberg et al. extend this analogy and introduce the notion of the $\lambda$-weighted synthetic gradient, which is analogous to the $\lambda$-return in RL. However, $\lambda$-weighted synthetic gradients were only presented conceptually and it remained unclear whether they would be of practical benefits as they still require BPTT.

In this study, inspired by established RL theory, we make conceptual and experimental advancements on $\lambda$-weighted synthetic gradients. In particular, we propose an algorithm for learning synthetic gradients, accumulate $BP(\lambda)$, which mirrors the accumulate TD($\lambda$) algorithm in RL (Van Seijen et al., 2016). Just as

how accumulate TD($\lambda$) provides an online solution to learning the $\lambda$-return in RL, we show that accumulate BP($\lambda$) provides an online solution to learning $\lambda$-weighted synthetic gradients. The algorithm uses forward-propagating eligibility traces and has the advantage of not requiring (even truncated) BPTT at all. Moreover, we demonstrate that accumulate BP($\lambda$) can alleviate the bias involved in directly learning bootstrapped estimations as suffered in the original implementation.

We now provide a brief background into the application of synthetic gradients for RNN learning. We then introduce the accumulate BP($\lambda$) algorithm and demonstrate both analytically and empirically can it alleviates the problem of bias suffered in the original implementation. Next, we touch upon accumulate BP($\lambda$) as a mechanism for learning in biological circuits. Finally, we discuss the limitations and conclusions of our work.

## 2  Background

### 2.1  Synthetic gradients for supervised learning

Consider an RNN with free parameters $\Psi$ performing a task of sequence length $T$ (which may be arbitrarily long). At time $t$ RNN dynamics follow $h_t = f(x_t, h_{t-1}; \Psi)$, where $h$ is the RNN hidden state, $x$ is the input, and $f$ is the RNN computation. Let $L_t = \mathcal{L}(\hat{y}_t, y_t)$ denote the loss at time $t$, where $\hat{y}_t$ is the RNN-dependent prediction and $y_t$ is the desired target. Let $L_{>t} = \sum_{t < \tau \leq T} L_\tau$ denote the total loss strictly after timestep $t$.

During training, we wish to update $\Psi$ to minimise all losses from timestep $t$ onwards. Using gradient descent this is achieved with $\Psi = \Psi - \eta \frac{\partial \sum_{t \leq \tau \leq T} L_\tau}{\partial \Psi}$ for some RNN learning rate $\eta$. We can write this gradient as

$$\frac{\partial \sum_{t \leq \tau \leq T} L_\tau}{\partial \Psi} = \frac{\partial (L_t + L_{>t})}{\partial \Psi} \tag{1}$$

$$= \frac{\partial L_t}{\partial \Psi} + \frac{\partial L_{>t}}{\partial \Psi} \tag{2}$$

$$= \left( \frac{\partial L_t}{\partial h_t} + \frac{\partial L_{>t}}{\partial h_t} \right) \frac{\partial h_t}{\partial \Psi} \tag{3}$$

Whilst the terms $\frac{\partial L_t}{\partial h_t}$ and $\frac{\partial h_t}{\partial \Psi}$ above are relatively easy to compute and available at timestep $t$, $\frac{\partial L_{>t}}{\partial h_t}$ can present challenges. Without BPTT, this term is simply taken as zero, $\frac{\partial L_{>t}}{\partial h_t} = 0$, and future errors are effectively ignored. With BPTT, this term is only computed after the forward pass and the corresponding backward pass so that all future errors are observed and appropriately backpropagated. This has memory and computational complexity which scales with $T$, and thereby relies on arbitrarily long waits before the loss gradient is available.

The aim of synthetic gradients is to provide an immediate prediction $\hat{G}_t$ of the future loss gradient, $\hat{G}_t \approx G_t := \frac{\partial L_{>t}}{\partial h_t}$ (Jaderberg et al., 2017). We use notation $G_t$ both to represent "gradient" but also to highlight its resemblance to the return in RL. Note that this gradient is a vector of the same size as $h$. As in the original implementation, we consider $\hat{G}_t$ to be a computation of the current RNN state with a separate "synthesiser" network: $\hat{G}_t = g(h_t; \theta)$, where $g$ denotes the synthesiser computation which depends on its free parameters $\theta$. An approximation for the loss gradient with respect to the RNN parameters can then be written as

$$\frac{\partial \sum_{t \leq \tau \leq T} L_\tau}{\partial \Psi} \approx \left( \frac{\partial L_t}{\partial h_t} + \hat{G}_t \right) \frac{\partial h_t}{\partial \Psi} \tag{4}$$

This is available at timestep $t$ and removes the dependence of the memory and computational complexity on $T$ as in Eq. 3.

## 2.2 Learning synthetic gradients

How the synthesiser parameters $\theta$ should be learned remains relatively unexplored and is the focus of this paper. The problem can be stated as trying to minimise the synthesiser loss function $L^g(\theta)$ defined as

$$L^g(\theta) := \mathbb{E}_{h_t \sim P} \left[ \frac{1}{2} \left\| v_t - \hat{G}_t \right\|_2^2 \right] \tag{5}$$

$$= \frac{1}{2} \sum_t P(h_t) \| v_t - g(h_t; \theta) \|_2^2 \tag{6}$$

where $P$ is the probability distribution over RNN hidden states, $\bar{G}_t$ is the target synthetic gradient for state $h_t$, and $\|.\|_2$ denotes the Euclidean norm. By taking samples of hidden states $h_t$ and applying the chain rule, the stochastic updates for $\theta$ can be written as

$$\theta_{t+1} = \theta_t + \alpha \left( v_t - g(h_t; \theta_t) \right)^\top \frac{\partial g(h_t; \theta_t)}{\partial \theta_t} \tag{7}$$

where $\alpha$ is the synthesiser learning rate. Note the transpose operation $\cdot^\top$ since all terms in Equation 7 are vectors.

Ideally $\bar{G}_t$ is the true error gradient, $\bar{G}_t = G_t$, but this requires full BPTT which is exactly what synthetic gradients are used to avoid. In its original formulation, Jaderberg et al. instead propose a target which is based on mixture of backpropagated error gradients within a fixed window and a bootstrapped synthetic gradient to incorporate errors beyond. The simplest version of this, which only uses one-step BPTT, only incorporates the error at the next timestep and relies on a bootstrap prediction for all timesteps onwards. We denote this target $G_t^{(1)}$.

$$G_t^{(1)} = \frac{\partial L_{t+1}}{\partial h_t} + \gamma \hat{G}_{t+1} \frac{\partial h_{t+1}}{\partial h_t} \tag{8}$$

where, inspired by RL, $\gamma \in [0, 1]$ is the gradient *discount factor* which if $\gamma < 1$ scales down later error gradients. Note that in its original implementation $\gamma = 1$, but in our simulations we find it important to set $\gamma < 1$ (see Appendix section C).

Notably, Equation 8 resembles the bootstrapped target involved in the TD algorithm in RL Sutton & Barto (2018). In particular, $G_t^{(1)}$ can be considered analogous to the one-step return at state $S_t$ defined as $R_{t+1} + \gamma V(S_{t+1})$, where $R_{t+1}$ is the reward, $S_{t+1}$ is the subsequent state, and $V$ is the (bootstrapped) value function.

Table 1: **Summary of terms used in value estimation in Reinforcement Learning (RL) and synthetic gradient (SG) estimation in supervised learning**. For RL $R_t$ denotes the reward at timestep $t$, $\phi_t$ denotes the feature-based representation of the environmental state, and $V$ is the value function. See main text and Van Seijen et al. (2016) for further details of the terms.

| | Reinforcement Learning | Synthetic Gradients |
|---|---|---|
| $G_t$ | $\sum_{\tau \geq 1} \gamma^{\tau-1} R_{t+\tau}$ | $\sum_{\tau \geq 1} \gamma^{\tau-1} \frac{\partial L_{t+\tau}}{\partial h_t}$ |
| $\hat{G}_t$ | $V(\phi_t; \theta)$ | $g(h_t; \theta)$ |
| $G_t^{(n)}$ | $\sum_{\tau=1}^n \gamma^{\tau-1} R_{t+\tau} + \gamma^n \hat{G}_{t+n}$ | $\sum_{\tau=1}^n \gamma^{\tau-1} \frac{\partial L_{t+\tau}}{\partial h_t} + \gamma^n \hat{G}_{t+n}^\top \frac{\partial h_{t+n}}{\partial h_t}$ |
| $G_t^\lambda$ | $(1-\lambda) \sum_{n=1}^{T-t-1} \lambda^{n-1} G_t^{(n)} + \lambda^{T-t-1} G_t$ | |
| $G_k^{\lambda|H}$ | $(1-\lambda) \sum_{n=1}^{H-k-1} \lambda^{n-1} G_k^{(n)} + \lambda^{H-k-1} G_t^{(H-k)}$ | |

## 2.3 $n$-step synthetic gradients

In practice, in its original implementation Jaderberg et al. primarily consider applying synthetic gradients alongside truncated BPTT for truncation size $n > 1$.

In this case, the left side term in Equation 8 can be extended to incorporate loss gradients backpropagated within the $n$ timesteps of the truncation. The synthesiser target is then set as $\bar{G}_t = G_t^{(n)}$ where $G_t^{(n)}$ is the *n-step synthetic gradient* defined as

$$G_t^{(n)} = \sum_{t < \tau \leq t+n} \gamma^{\tau-t-1} \frac{\partial L_\tau}{\partial h_t} + \gamma^n \hat{G}_{t+n} \frac{\partial h_{t+n}}{\partial h_t} \tag{9}$$

which is analogous to the $n$-step return in RL. Importantly, in practice this scheme uses truncated BPTT to both define the synthesiser target and the error gradient used to update the RNN. Specifically, Jaderberg et al. apply the $n$ backpropagated gradients to directly learn $\Psi$ as well as $\theta$, with the synthetic gradients themselves only applied at the end of each truncation window (see original paper for details).

Just as in RL, increasing the truncation size $n$ provides a target which assigns more weight to the observations than the bootstrapped term, thus reducing its *bias* and potentially leading to better synthesiser learning. On the other hand, Equation 9 requires $n$-step BPTT and thereby enforces undesirable waiting time and complexity for large $n$.

## 3 Model and analytical results

In this study we formulate a learning algorithm – accumulate BP($\lambda$) – which has the advantage of reduced bias compared to the original $n$-step implementation. Furthermore, the algorithm can be implemented at each timestep and does not require BPTT at all.

We first define $\lambda$-weighted synthetic gradients. We highlight that our definition is similar, but not the same, as that first introduced in Jaderberg et al. (2017). Specifically, our definition incorporates loss gradients for losses strictly after the current timestep and can therefore naturally be used in the context of learning the synthesiser.

### 3.1 $\lambda$-weighted synthetic gradient

Let $\lambda$ be such that $0 \leq \lambda \leq 1$. We define the $\lambda$-*weighted synthetic gradient* as

$$G_t^\lambda := (1 - \lambda) \sum_{n=1}^{\infty} \lambda^{n-1} G_t^{(n)} \tag{10}$$

$$= (1 - \lambda) \sum_{n=1}^{T-t-1} \lambda^{n-1} G_t^{(n)} + \lambda^{T-t-1} G_t \tag{11}$$

which is analogous to the $\lambda$-return in RL.

Note the distinction in notation between the $\lambda$-weighted synthetic gradient $G_t^\lambda$ and the $n$-step synthetic gradient $G_t^{(n)}$. Moreover, note that Equation 11 is similar but distinct to the recursive definition as proposed in the original paper (see Appendix section B). In particular, whilst Jaderberg et al. also incorporate the loss gradient at the current timestep in their definition, Equation 11 only considers strictly future losses. Since the synthesiser itself is optimised to produce future error gradients (Equation 4), this enables the $\lambda$-weighted synthetic gradient to be directly used as a synthesiser target.

As in RL, a higher choice of $\lambda$ results in stronger weighting of observed gradients compared to the bootstrapped terms. For example, whilst $G_t^0 = G_t^{(1)}$ is just the one-step synthetic gradient which relies strongly on the bootstrapped prediction (cf. Equation 8), $G_t^1 = G_t$ is the true (unbiased) gradient as obtained via full BPTT.

We also define the *interim $\lambda$-weighted synthetic gradient* $G_k^{\lambda|H}$ as

$$G_k^{\lambda|H} := (1 - \lambda) \sum_{n=1}^{H-k-1} \lambda^{n-1} G_k^{(n)} + \lambda^{H-k-1} G_k^{(H-k)} \tag{12}$$

---

**Algorithm 1 RNN learning with accumulate BP($\lambda$).** Updates RNN parameters using estimated gradients provided by synthesiser function $g$.

---

**Input:** $\Psi_0$, $\theta_0$, $\{(x_t, y_t)\}_{1 \leq t \leq T}$, $\eta$, $\alpha$, $\gamma$, $\lambda$

   $\Psi \leftarrow \Psi_0$                                                {init. RNN parameters}

   $\theta \leftarrow \theta_0$                                                {init. synthesiser parameters}

   $h, \partial_h, e \leftarrow 0$                                   {init. RNN state, Jacobian, and elig. trace}

   **for** $t = 1$ **to** $T$ **do**

      $e \leftarrow \gamma\lambda\partial_h e + \frac{\partial g(h;\theta)}{\partial \theta}$                    {update eligibility trace}

      $h' \leftarrow f(x_t, h; \Psi)$, $L \leftarrow \mathcal{L}(h', y_t)$       {compute next hidden state and task loss}

      $\partial_h \leftarrow \frac{\partial h'}{\partial h}$, $\partial_L \leftarrow \frac{\partial L}{\partial h'}$, $\partial_\Psi \leftarrow \frac{\partial h'}{\partial \Psi}$      {compute local gradients}

      $\delta \leftarrow [\partial_L + \gamma g(h'; \theta)]^\top \partial_h - g(h; \theta)$    {compute synthesiser TD error}

      $\Delta\theta \leftarrow \alpha \delta^\top e$                               {update synthesiser parameters}

      $\Delta\Psi \leftarrow \eta[\partial_L + g(h'; \theta)]^\top \partial_\Psi$         {update RNN parameters}

      $h \leftarrow h'$                                   {update RNN hidden state}

   **end for**

---

Unlike Equation 11, this is available at time ("horizon") $H$ with $k < H < T$.

Table 1 provides an overview of the terms defined along with their respective counterparts in RL.

### 3.2 Offline $\lambda$-**SG algorithm**

We define the *offline $\lambda$-SG algorithm* to learn $\theta$, which is analogous to the offline $\lambda$-return algorithm in RL but for synthetic gradients (SG), by taking $\bar{G}_t = G_t^\lambda$ in Equation 7. It is offline in the sense that it requires the completion of the sequence at timestep $T$ before updates are possible.

### 3.3 Online $\lambda$-**SG algorithm**

We define the *online $\lambda$-SG algorithm* to learn $\theta$, which is analogous to the online $\lambda$-return algorithm in RL but for synthetic gradients. At the current timestep $t$, the algorithm updates $\theta$ based on its prediction over all prior RNN hidden states $h_k$ from $k = 0$ up to $k = t$. In particular, at timestep $t$ the algorithm applies as many weight updates by iterating over $k$ according to

$$\theta_k^t = \theta_{k-1}^t + \alpha \left( G_k^{\lambda|t} - g(h_k; \theta_{k-1}^t) \right)^\top \frac{\partial g(h_k; \theta_{k-1}^t)}{\partial \theta_{k-1}^t} \tag{13}$$

where $\theta_0^0 = \theta_{\text{init}}$ is the initialisation weight and subsequent initial weight vectors are inherited from the previous timestep, $\theta_0^{t+1} = \theta_t^t$. This latter quantity – the final weight vector at time $t$ – defines $\theta_t$ (without superscript).

The information used for the weight updates in Equation 13 is available at the current timestep and the algorithm is therefore online. Moreover, as in RL, the online $\lambda$-SG algorithm produces weight updates similar to the offline $\lambda$-SG algorithm. In particular, at the end of the sequence with the horizon $H = T$ note that $G_t^{\lambda|H}$ and $G_t^\lambda$ are the same. However, to store the previous hidden states and iteratively apply the online $\lambda$-SG algorithm requires undesirable computational cost. In particular, the $1 + 2 + \cdots + T$ operations in Equation 13 result in computational complexity which scales intractably with $T^2$.

### 3.4 Accumulate BP($\lambda$)

In this study we propose the accumulate BP($\lambda$) algorithm which is directly inspired by the accumulate TD($\lambda$) algorithm in RL (Van Seijen et al., 2016). Like accumulate TD($\lambda$), the motivation for accumulate BP($\lambda$) is to enable relatively cheap, online computations whilst alleviating the problem of bias which comes from bootstrapping using *eligibility traces* (Fig. 1**b**).

In accumulate BP($\lambda$), the weight update at timestep $t$ is defined by

$$\theta_{t+1} = \theta_t + \alpha \delta_t^\top e_t \tag{14}$$

Where $\delta_t$ is the temporal difference error at timestep $t$

$$\delta_t = \frac{\partial L_{t+1}}{\partial h_t} + \gamma g(h_{t+1}; \theta_t)^\top \frac{\partial h_{t+1}}{\partial h_t} - g(h_t; \theta_t) \tag{15}$$

and $e_t$ is the eligibility trace of $\theta$ at time $t$

$$e_t = \gamma \lambda \frac{\partial h_t}{\partial h_{t-1}} e_{t-1} + \frac{\partial g(h_t; \theta_t)}{\partial \theta_t} \tag{16}$$

with $e_0$ defined as the zero vector, $e_0 = 0$.

**Point of clarity**. Note that there is some abuse of notation with respect to the matrix multiplication operations defined in Equations 14 and 16. If $\text{in}_\theta, \text{out}_\theta$ are the sizes of the input and output dimension of $\theta$, respectively, then the eligibility trace $e_t$ is a three-dimensional vector of shape $(|h|, \text{out}_\theta, \text{in}_\theta)$, where $|h|$ is size of $h$. To compute the matrix product $A e_t$ for a vector $A$ of shape $(r, |h|)$ we concatenate the latter two dimensions of $e_t$ so that is of shape $(|h|, \text{out}_\theta \times \text{in}_\theta)$ and once the product is computed reshape it as $(r, \text{out}_\theta, \text{in}_\theta)$. Note that if $r = 1$ (as in Equation 14) then the first dimension is removed.

The core idea behind accumulate BP($\lambda$) is that these eligibility traces keep track of the weights which have contributed to previous error-gradient predictions, where the rate at which this contribution fades over time is $\lambda\gamma$. These traces then inform the synthesiser which weights are more liable to change once a temporal difference error occurs (Equation 15). Importantly, accumulate BP($\lambda$) applies a forward-view (online) method of learning which operates timestep by timestep, in stark contrast to the (backward-view) offline $\lambda$-SG algorithm for which the task sequence must first be completed before updates can occur.

The main analytical result in this paper is that accumulate BP($\lambda$) provides an approximation to the online $\lambda$-SG algorithm. This theorem uses the term $\Delta_i^t$ defined as

$$\Delta_i^t := \left( G_{i,0}^{\lambda|t} - g(h_i; \theta_0) \right)^\top \frac{\partial g(h_i; \theta_i)}{\partial \theta_i} \tag{17}$$

where $G_{i,0}^{\lambda|t}$ is the $\lambda$-weighted synthetic gradient which uses the initial weight vector $\theta_0$ for all synthetic gradient estimations.

**Theorem 3.1.** *Let $\theta_0$ be the initial weight vector, $\theta_t^{BP}$ be the weight vector at time $t$ computed by accumulate BP($\lambda$), and $\theta_t^\lambda$ be the weight vector at time $t$ computed by the online $\lambda$-SG algorithm. Furthermore, assume that $\sum_{i=0}^{t-1} \Delta_i^t$ does not contain any zero-elements. Then, for all time steps $t$:*

$$\frac{\left\| \theta_t^{BP} - \theta_t^\lambda \right\|_2}{\left\| \theta_t^{BP} - \theta_0 \right\|_2} \to 0 \quad as \quad \alpha \to 0$$

**Proof:** The full proof can be found in Appendix section A. In general, the structure of the proof closely follows that provided in the analogous RL paradigm for accumulate $TD(\lambda)$ (Van Seijen et al., 2016) $\square$.

Accumulate BP($\lambda$) thus provides an online method for learning the $\lambda$-weighted synthetic gradient which, analogous to accumulate $TD(\lambda)$, avoids the memory and computational requirements of the online $\lambda$-SG algorithm. For example, when eligibility traces are unused and $\lambda = 0$, i.e. accumulate $BP(0)$, the synthesiser learns with the one-step synthetic gradient as its target, $\bar{G}_t = G_t^{(1)}$, and arrives at the original implementation with truncation size $n = 1$ (Jaderberg et al., 2017). When $\lambda = 1$, i.e. accumulate $BP(1)$, for an appropriately small learning rate $\alpha$ the synthesiser learns with true BPTT gradient as its target, $\bar{G}_t = G_t$. Importantly, Equations 15 and 16 only consider gradients between variables of at most 1 timestep apart. In this respect, at least in the conventional sense, there is no BPTT.

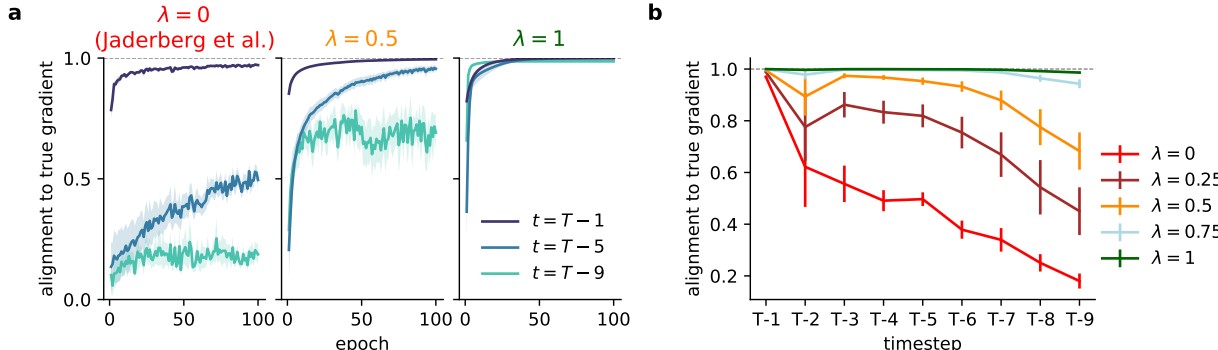

Figure 2: **BP($\lambda$) derives true BPTT gradients over multiple timesteps.** In this toy paradigm, input is only provided at timestep 1 and the task target is only available at the end of the task at time $T = 10$. (**a**) Alignment between synthetic gradients and true gradients for a fixed RNN model across different timesteps within the task, where the synthetic gradients are learned using (accumulate) BP($\lambda$). Alignment is defined using the cosine similarity metric. (**b**) The average alignment over the last 10% of epochs in **a** across all timesteps.

## 4 Experiments

Next, we tested empirically the ability of accumulate BP($\lambda$), which we henceforth simply call BP($\lambda$), to produce good synthetic gradients and can drive effective RNN parameter updates. In all experiments we take the synthesiser computation $g$ simply as a linear function of the RNN hidden state, $g(h_t; \theta) = \theta h_t$.

### 4.1 Approximating true error gradients in a toy task

We first analyse the alignment of BP($\lambda$)-derived synthetic gradients and true gradients derived by full BPTT, which we use to quantify the bias in synthesiser predictions. For this we consider a toy task in which a fixed (randomly connected) linear RNN receives a static input $x_1$ at timestep 1 and null input onwards, $x_t = 0$ for $t > 1$. To test the ability of BP($\lambda$) to transfer error information across time the error is only defined at the last timestep $L_T$, where $L_T$ is the mean-squared error (MSE) between a two-dimensional target $y_T$ and a linear readout of the final hidden activity $h_T$. We use a task length of $T = 10$ timesteps. Note that since the network is linear and the loss is a function of the MSE, the linear synthesiser should in principle be able to learn perfectly model the true BPTT gradient (Czarnecki et al., 2017).

As expected, we find that a high $\lambda$ improves the alignment of synthetic gradients and true gradients compared to the $\lambda = 0$ case (i.e. BP(0) as in Jaderberg et al. (2017); Fig. 2**a**). Specifically, these results show, as predicted, that the heavy reliance on bootstrapping of BP(0) means that loss gradients near the end of the sequence must first be faithfully captured before earlier timesteps can be learned. When eligibility traces are applied in BP(1), however, the over-reliance on the bootstrapped estimate is drastically reduced and the synthesiser can very quickly learn faithful predictions across all timesteps. Indeed, we observe that high $\lambda$ avoids the deterioration of synthetic gradient quality at earlier timesteps as suffered by BP(0) (Fig. 2**b**). We also observe that BP($\lambda$) often outperforms $n$-step synthetic gradients (Fig. 6).

Next, to verify that BP($\lambda$) is actually beneficial for RNN learning, we followed the scheme of Jaderberg et al. (2017) and applied RNN weight updates together with synthesiser weight updates (Algorithm 1). To ensure that the RNN needs to learn a temporal association (as opposed to a fixed readout), in this case we consider 3 input/target pairs and consider large sequence lengths $T$; we also now apply a tanh non-linearity to bound RNN activity.

Consistent with our results for the case of fixed RNNs, we observe that BP($\lambda$) gives rise to better predictions for high $\lambda$ when the RNN is learning (Fig. 3**a**). Notably, however, the alignment is weaker for plastic RNNs

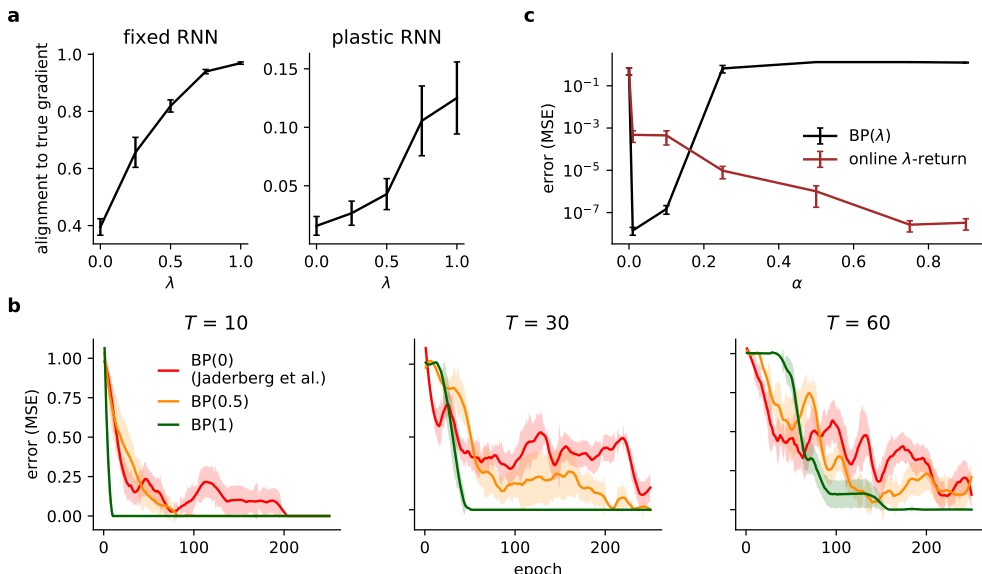

Figure 3: **BP(λ) drives better RNN learning in a toy task**. (**a**) Average cosine similarity between synthetic gradients and true gradients for fixed (left) and plastic (right) RNNs. Cosine similarity for plastic RNNs is taken over the first 5 training epochs, since this initial period is the key stage of learning (i.e. before the task is perfected). (**b**) Learning curves of RNNs which are updated using synthetic gradients derived by BP(λ) over different sequence lengths $T$. (**c**) Effect of learning rate $\alpha$ (cf. Equation 14) on task error (averaged over last 10 epochs of training) for plastic RNNs using synthetic gradients derived by BP(λ) and the online λ-SG algorithm ($\lambda = 1$ $T = 10$); this can be compared to Figure 2 in Van Seijen et al. (2016) which displays analagous results in the RL paradigm. Here standard stochastic gradient descent is used (all other main experiments use Adam). Results show average (with SEM) over 5 different initial conditions.

when compared to fixed RNNs. This is because the challenge is now harder for the synthesiser, since changes to the RNN will affect its error gradients and therefore lead to a moving synthesiser target. Nonetheless, we find that BP(λ) improves RNN learning even in the presence of relatively long temporal credit assignment (Fig. 3**b**). Next, we contrasted the ability of both BP(λ) and $n$-step methods to achieve near-zero error. Our results show that BP(1) is able to solve tasks more than double the length of those solved by the next best model (Table 2 and Fig. 7).

Finally, we explored the effect of learning rate with BP(λ). Indeed, Theorem 3.1 only applies as the step-size $\alpha$ approaches zero, and empirically TD(λ) is known to diverge if $\alpha$ is too large Van Seijen et al. (2016). We observe similar results in our experiments, and note that BP(λ) is only applicable with modest step-sizes (Fig. 3**c**). This is true over a range of values for λ, though, consistent with TD(λ), we observe that mid-range λ values are more robust to the learning rate (Fig. 8; e.g. compare $\lambda = 0.75$ with $\lambda = 1$).

## 4.2 Sequential MNIST task

To test the ability of BP(λ) to generalise to non-trivial tasks, we now consider the sequential MNIST task (Le et al., 2015). In this task the RNN is provided with a row-by-row representation of an MNIST image which it must classify at the end (Fig. 4**a**). That is, as in the toy task above, the task loss is only defined at the final timestep and must be effectively associated to prior inputs. Since this is a harder task we now use non-linear LSTM units in the RNN which are better placed for these temporal tasks (Hochreiter & Schmidhuber, 1997).

Table 2: **Overview of model performance for sequential MNIST and copy-repeat tasks**. Results for toy and copy-repeat tasks shows the average task sequence length solved by the models (see text). Results for the sequential MNIST task show the average test accuracy as a percentage after training. Values denote average over 5 different initial conditions.

| | BPTT | | | | BPTT + SG | | | | no BPTT | | | |
| --- | --- | --- | --- | --- | --- | --- | --- | --- | --- | --- | --- | --- |
| | n=2 | n=3 | n=4 | n=5 | n=2 | n=3 | n=4 | n=5 | n=1 | BP(0) | BP(0.5) | BP(1) |
| toy task | 16 | 10 | 20 | 8 | 36 | 38 | 24 | 20 | 4 | 16 | 32 | **90** |
| sequential MNIST | 73.4 | 78.4 | 82.0 | 87.1 | 78.2 | 83.0 | 86.5 | 90.4 | 64.1 | 69.6 | 76.6 | **90.6** |
| copy-repeat | 9.0 | 9.0 | 12.0 | 15.0 | 12.0 | 9.0 | 15.0 | 23.0 | 8.2 | 9.0 | 15.8 | **29.0** |

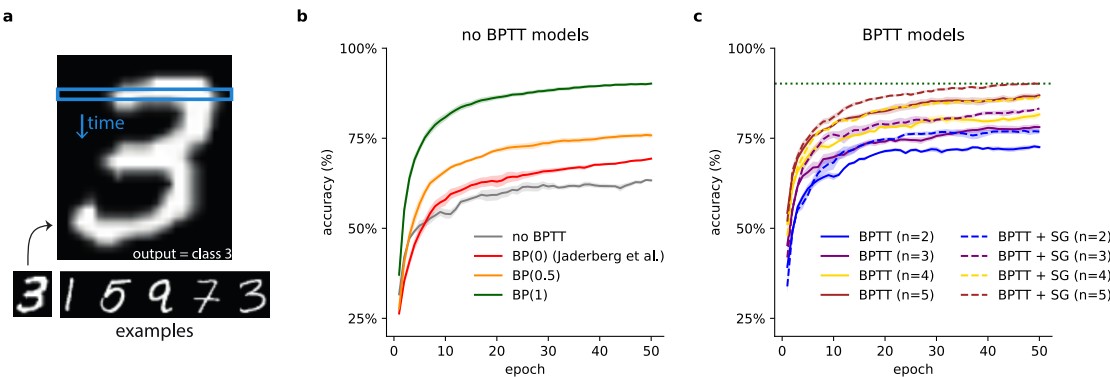

Figure 4: **Performance of BP($\lambda$) in sequential MNIST task**. (**a**) Schematic of task. Rows of an MNIST image are fed sequentially as input and the model must classify the digit at the end. (**b**) Validation accuracy during training for BP($\lambda$) models. (**c**) Validation accuracy during training for models which learn synthetic gradients (SG) with $n$-step truncated BPTT as in original implementation (Jaderberg et al., 2017); final performance of BP(1) (as in (b); dotted green) is given for reference. Results show mean performance over 5 different initial conditions with shaded areas representing standard error of the mean.

Our results show that the use of BP($\lambda$) significantly improves on a standard no-BPTT model, i.e. with $\frac{\partial L_{>t}}{\partial h_t} = 0$ in Eq. 3 (Figs. 4**b** and 9). Consistent with our results from the toy task we find that a high $\lambda$ produces faster rates of learning and higher performance. For example, BP(1) achieves error less than a third of that achieved by the eligibility trace free model, BP(0) ($\sim 10\%$ vs $\sim 30\%$). Moreover, BP(1) generally outperforms the models considered by Jaderberg et al. (2017) which rely on truncated BPTT and the $n$-step synthetic gradient with $n > 1$ (Fig. 4**c** and Table 2).

### 4.3 Copy-repeat task

Finally, we consider a task with more complex and longer temporal dependencies – the copy-repeat task (Graves et al., 2014). In this task the model receives as input a start delimiter followed by an 8-dimensional binary sequence of length $N$ and a repeat character $R$. The model must then output the sequence $R$ times before finishing with a stop character. The total sequence length is then $T = N \times (R + 1) + 3$. We follow the procedure as set out in Jaderberg et al. (2017) and deem a sequence length solved if the average error is less than 0.15 bits. Once solved we increment $N$ and $R$ alternatively.

We again observe that BP(1) provides the best BPTT-free solution in solving large sequence lengths and outperforms the $n$-step synthetic gradient methods (Fig. 5 and Table. 2). We do however note that our implementation of these $n$-step methods fails to reach the performance thresholds as presented in the original

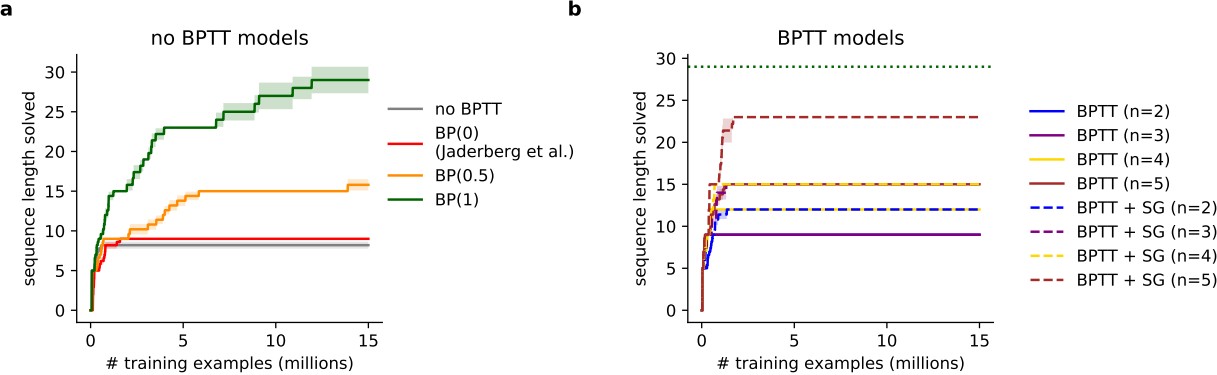

Figure 5: **Performance of** BP($\lambda$) **in copy-repeat task**. (**a**) Maximum sequence length solved for BP($\lambda$) models. A sequence length is considered as solved if the model achieves an average of 0.15 bits error for a given length. (**b**) Maximum sequence length solved for models with $n$-step synthetic gradient (SG) learning methods (Jaderberg et al., 2017); best task performance of BP(1) (as in (a); dotted green) is shown for reference. See also Table 2 for more details. Results show mean performance over 5 different initial conditions with shaded areas representing standard error of the mean.

paper (Jaderberg et al., 2017), but that they are consistent with those reported in Bellec et al. (2019). This is perhaps due to the more modest hyperparameter choices (e.g. less hidden units) used in our models.

## 5 Relevance for learning in biological networks

Table 3: **Complexity of the BP($\lambda$) algorithm against backpropagation through time (BPTT) and real-time recurrent learning (RTRL) based algorithms for fully connected recurrent neural networks (RNNs)**. $|h|$ is the hidden size of the RNN; $T$ is the task length; $n$ is the imposed truncation size. Time complexity is reported per timestep (i.e. the average complexity over the sequence of timesteps) and per "pass", where pass refers to the period over which the gradient is computed (e.g. over each truncation for truncated BPTT, or over each timestep for RTRL and BP($\lambda$)). $|\Psi|$ denotes the number of parameters in the RNN, whilst $|\theta|$ denotes the number of parameters in the synthesiser.

| Algorithm | Memory | Time (per timestep) | Time (per pass) |
|---|---|---|---|
| Full BPTT | $|h|T$ | $|h|^2$ | $|h|^2T$ |
| Truncated BPTT | $|h|n$ | $|h|^2$ | $|h|^2n$ |
| Synthetic gradients (original) | $|h|n$ | $|h|^2$ | $|h|^2n$ |
| **BP($\lambda$) (ours)** | $|h||\theta|$ | $|h|^2|\theta|$ | $|h|^2|\theta|$ |
| RTRL | $|h||\Psi|$ | $|h|^2|\Psi|$ | $|h|^2|\Psi|$ |

The BP($\lambda$) algorithm is of potential interest to neuroscience. Unlike BPTT which is considered biologically implausible (Lillicrap & Santoro, 2019; Prince et al., 2021), BP($\lambda$) is fully online and avoids the need to store intermediate activations (and complex gradient calculations back in time) across an arbitrary number of timesteps. Moreover, the application of synthetic gradients has relatively cheap computational and memory costs when compared to other online learning algorithms (Marschall et al., 2020). Perhaps most interestingly, BP($\lambda$) employs a combination of retrospective and prospective learning, each of which are thought to take place in the brain (Namboodiri & Stuber, 2021). Specifically, whilst the main network learns via prospective learning signals (i.e. synthetic gradients), the synthesiser itself learns in a retrospective manner by using forward-propagating eligibility traces.

One possible candidate for the expression of synthetic gradients in the nervous system are neuromodulators. For example, dopaminergic neurons are known to encode expectation of future reward or error signals (Hollerman & Schultz, 1998) and have been observed to play an important role in mediating synaptic plasticity (Yagishita et al., 2014; Gerstner et al., 2018). Furthermore, as required for synthetic gradient vectors, there is increasing evidence for significant heterogeneity in the dopaminergic population in areas such as the ventral tegmental area, both in its variety of encoded signals and the targeted downstream circuits (Lerner et al., 2015; Beier et al., 2015; Avvisati et al., 2022). Each signal may thus reflect a predicted gradient with respect to the target cortical cell or cell ensemble.

More recently, it has also been suggested that a particular subcortical structure – the cerebellum – predicts cortical error gradients via the cortico-cerebellar loop (Pemberton et al., 2021; Boven et al., 2023). The cerebellum would thus act as a synthesiser for the brain, and it is suggested that a bootstrapped learning strategy may be in line with experimentally observations (Ohmae S, 2019; Kawato et al., 2021). BP($\lambda$) offers an extra degree of biological plausibility to these studies by removing the need for BPTT at all. Moreover, the algorithm makes specific predictions regarding the need for eligibility traces (cf. Eq. 16) at key learning sites such as the cerebellar parallel fibres (Kawato et al., 2011).

## 6 Limitations

Like the originally proposed algorithm for synthetic gradients, BP($\lambda$) makes the assumption that future error gradients can be modeled as a function of the current activity in the task-performing network. The algorithm may therefore suffer when the mapping between activity and future errors becomes less predictable, for example when the task involves stochastic inputs. Indeed, we find that whilst BP($\lambda$) works well for tasks with fairly structured temporal structure as in the experiments presented, the algorithm can fall short in tasks with little or no temporal correlation between task inputs. For example, we failed to observe meaningful gains with BP($\lambda$) in the LISTOPS task which employs randomly generated inputs (Tay et al., 2020). On a related note, it may be that in the case of variable, non-deterministic settings BP($\lambda$) may be prone to a high variance in the synthesiser target gradient compared to the more biased BP(0), resulting in potential instability during learning as can occur, for example, in Monte Carlo algorithms in RL. However, how the bias-variance tradeoff in RL exactly translates to BP($\lambda$) is not trivial. This is because in RL the biases and variances are computed on a more linear relationship between states and values, whereas in our case gradients have complex dependencies induced by the RNN Jacobian (cf. Eq. 15. Indeed, when we tried to directly analyse the bias and variance of the gradient vector, we were unable to obtain meaningful results. Instead in our results we simply quantify the goodness-of-fit by the cosine similarity metric and the overall RNN performance. In this case a high $\lambda$ is consistently the best choice, but whether mid-range values can be beneficial for certain task conditions, as can happen in RL, remains to be tested in future work.

Additionally, whilst the complexity of BP($\lambda$) does not depend on time, the algorithm is still expensive, particularly if there are many hidden units in the RNN (Table 3). For example, if there are $|h|$ RNN units and a linear synthesiser is applied then, due to the requirements of storing and updating the synthesiser eligibility traces, the memory and computational complexity of the algorithm is $\mathcal{O}(|h|^3)$ and $\mathcal{O}(|h|^4)$ respectively (cf. Table 3); this is as costly as the notoriously expensive real-time recurrent learning algorithm (Williams & Zipser, 1989). We postulate that one way to alleviate this issue is to not learn error gradients with respect to RNN activity directly, but instead some low dimensional representation of that error gradient. In particular, it has been recently demonstrated that gradient descent often takes place within a small subspace Gur-Ari et al. (2018). In principle, therefore, the synthesiser could learn some encoding of the error gradient of dimensionality $s << |h|$, significantly reducing the complexity of BP($\lambda$). We predict that such low dimensional representations, which can reduce the noise of high dimensional error gradients, may also lead to more stable synthesiser learning. Speculating further, it may be that this bottleneck role is played by the thalamus in the communication of predicted error feedback in cerebellar-cortico loops (see previous section; Pemberton et al. (2021)).

Finally, we highlight that BP($\lambda$) requires the dynamics of the main model to be represented by sequentially dependent states; i.e. the Jacobian $\frac{\partial h_{t+1}}{\partial h_t} \neq 0$, as is inherently satisfied by RNNs. However, we note that BP($\lambda$) may in principle be applied to feedforward networks where activities are now defined across layers

instead of timesteps – similar to what was done by Jaderberg et al. (2017). However, according to our formulation BP($\lambda$) the same synthesiser parameters should operate across layers, thus future work should explore how to best relax this constraint.

## 7 Conclusion

BPTT can be expensive and enforce long waiting times before gradients become available. Synthetic gradients remove these locking constraints imposed by BPTT as well as the associated computational and memory complexity with respect to the task length (Jaderberg et al., 2017). However, the bootstrapped $n$-step algorithm for learning synthetic gradients as proposed by Jaderberg et al. can lead to biased estimates and also maintains some dependence on (truncated) BPTT to be performed.

Inspired by the TD($\lambda$) algorithm in RL we propose a novel algorithm for learning synthetic gradients: BP($\lambda$). This algorithm applies forward propagating eligibility traces in order to reduce the bias of its estimates and is fully online. We thus extend the work of Jaderberg et al. in developing a computational bridge between estimating expected return in the RL paradigm and future error gradients in supervised learning.

Through a combination of analytical and empirical work we show that BP($\lambda$) outperforms the original implementation for synthetic gradients. Moreover, our model offers an efficient online solution for temporal supervised learning that is of relevance for both artificial and biological networks.

### Acknowledgments

We would like to thank the Neural & Machine Learning group and the Joao Sacramento group for useful feedback. J.P. was funded by a EPSRC Doctoral Training Partnership award (EP/R513179/1) and R.P.C. by the Medical Research Council (MR/X006107/1), BBSRC (BB/X013340/1) and a ERC-UKRI Frontier Research Guarantee Grant (EP/Y027841/1). This work made use of the HPC system Blue Pebble at the University of Bristol, UK. We would like to thank Dr Stewart for a donation that supported the purchase of GPU nodes embedded in the Blue Pebble HPC system.

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

# A Proof of Theorem 3.1

As this section is more involved, for clarity we denote scalar variables in italics and vectors in bold.

**Approach**: We demonstrate that Theorem 3.1 holds for arbitrary synthesiser parameter $\theta^{ji}$ (at the $i$th column and $j$th row of $\theta$) ; that is,

$$\frac{\left\| \theta_t^{ji,BP} - \theta_t^{ji,\lambda} \right\|_2}{\left\| \theta_t^{ji,BP} - \theta_0^{ji} \right\|_2} \to 0, \quad \text{as} \quad \alpha \to 0.$$

This is proved in Proposition A.7.

For reference, we explicitly write the weight updates for parameter $\theta^{ji}$ according to the accumulate BP($\lambda$) algorithm:

$$\boldsymbol{e}_t^{ji} = \gamma\lambda \frac{\partial \boldsymbol{h}_t}{\partial \boldsymbol{h}_{t-1}} \boldsymbol{e}_{t-1}^{ji} + \frac{\partial g(\boldsymbol{h}_t; \boldsymbol{\theta}_t)}{\partial \theta_t^{ji}}, \tag{18}$$

$$\boldsymbol{\delta}_t = \frac{\partial L_{t+1}}{\partial \boldsymbol{h}_t} + \gamma g(\boldsymbol{h}_{t+1}; \boldsymbol{\theta}_t)^\top \frac{\partial \boldsymbol{h}_{t+1}}{\partial \boldsymbol{h}_t} - g(\boldsymbol{h}_t; \boldsymbol{\theta}_t), \tag{19}$$

$$\theta_{t+1}^{ji} = \theta_t^{ji} + \alpha \boldsymbol{\delta}_t^\top \boldsymbol{e}_t^{ji}. \tag{20}$$

we also define the helper variable $\boldsymbol{\delta}_{a,t}$ as

$$\boldsymbol{\delta}_{a,t} = \frac{\partial L_{t+1}}{\partial \boldsymbol{h}_a} + \gamma g(\boldsymbol{h}_{t+1}; \boldsymbol{\theta}_t)^\top \frac{\partial \boldsymbol{h}_{t+1}}{\partial \boldsymbol{h}_a} - g(\boldsymbol{h}_t; \boldsymbol{\theta}_t)^\top \frac{\partial \boldsymbol{h}_t}{\partial \boldsymbol{h}_a} = \boldsymbol{\delta}_t^\top \frac{\partial \boldsymbol{h}_t}{\partial \boldsymbol{h}_a}. \tag{21}$$

In general, we follow the structure of the analogous proof for the accumulate TD($\lambda$) algorithm (Van Seijen et al., 2016).

## A.1 Statements and proofs

**Lemma A.1.** $\boldsymbol{G}_a^{\lambda|t+1} = \boldsymbol{G}_a^{\lambda|t} + (\lambda\gamma)^{t-a}\boldsymbol{\delta}_{a,t}'$, where

$$\boldsymbol{\delta}_{a,t}' = \frac{\partial L_{t+1}}{\partial \boldsymbol{h}_a} + \gamma g(\boldsymbol{h}_{t+1}; \boldsymbol{\theta}_t)^\top \frac{\partial \boldsymbol{h}_{t+1}}{\partial \boldsymbol{h}_a} - g(\boldsymbol{h}_t; \boldsymbol{\theta}_{t-1})^\top \frac{\partial \boldsymbol{h}_t}{\partial \boldsymbol{h}_a}. \tag{22}$$

**Proof**:

$$
\begin{aligned}
\boldsymbol{G}_a^{\lambda|t+1} - \boldsymbol{G}_a^{\lambda|t} =& (1-\lambda)\sum_{n=1}^{t-a}\lambda^{n-1}\boldsymbol{G}_a^{(n)} + \lambda^{t-a}\boldsymbol{G}_a^{(t+1-a)} && \text{Definition: Equation 12} \\
& - (1-\lambda)\sum_{n=1}^{t-a-1}\lambda^{n-1}\boldsymbol{G}_a^{(n)} - \lambda^{t-a-1}\boldsymbol{G}_a^{(t-a)} \\
=& (1-\lambda)\lambda^{t-a-1}\boldsymbol{G}_a^{(t-a)} + \lambda^{t-a}\boldsymbol{G}_a^{(t+1-a)} - \lambda^{t-a-1}\boldsymbol{G}_a^{(t-a)} \\
=& \lambda^{t-a}\boldsymbol{G}_a^{(t+1-a)} - \lambda^{t-a}\boldsymbol{G}_a^{(t-a)} \\
=& \lambda^{t-a}\left(\boldsymbol{G}_a^{(t+1-a)} - \boldsymbol{G}_a^{(t-a)}\right) \\
=& \lambda^{t-a}\left(\sum_{k=1}^{t+1-a}\gamma^{k-1}\frac{\partial L_{a+k}}{\partial \boldsymbol{h}_a} + \gamma^{t+1-a}g(\boldsymbol{h}_t;\boldsymbol{\theta}_t)^\top\frac{\partial \boldsymbol{h}_{t+1}}{\partial \boldsymbol{h}_a} - \sum_{k=1}^{t-a}\gamma^{k-1}\frac{\partial L_{a+k}}{\partial \boldsymbol{h}_a} - \gamma^{t-a}g(\boldsymbol{h}_t;\boldsymbol{\theta}_{t-1})^\top\frac{\partial \boldsymbol{h}_t}{\partial \boldsymbol{h}_a}\right) \\
=& \lambda^{t-a}\left(\gamma^{t-a}\frac{\partial L_{t+1}}{\partial \boldsymbol{h}_a} + \gamma^{t+1-a}g(\boldsymbol{h}_t;\boldsymbol{\theta}_t)^\top\frac{\partial \boldsymbol{h}_{t+1}}{\partial \boldsymbol{h}_a} - \gamma^{t-a}g(\boldsymbol{h}_t;\boldsymbol{\theta}_{t-1})^\top\frac{\partial \boldsymbol{h}_t}{\partial \boldsymbol{h}_a}\right) \\
=& (\lambda\gamma)^{t-a}\underbrace{\left(\frac{\partial L_{t+1}}{\partial \boldsymbol{h}_a} + \gamma g(\boldsymbol{h}_t;\boldsymbol{\theta}_t)^\top\frac{\partial \boldsymbol{h}_{t+1}}{\partial \boldsymbol{h}_a} - g(\boldsymbol{h}_t;\boldsymbol{\theta}_{t-1})^\top\frac{\partial \boldsymbol{h}_t}{\partial \boldsymbol{h}_a}\right)}_{\boldsymbol{\delta}'_{a,t}} && \square
\end{aligned}
$$

**Lemma A.2.** $\boldsymbol{G}_a^{\lambda|t} = \boldsymbol{G}_a^{\lambda|a+1} + \sum_{b=a+1}^{t-1}(\gamma\lambda)^{b-a}\boldsymbol{\delta}'_{a,b}$

**Proof:** We apply Lemma A.1 recursively:

$$
\begin{aligned}
\boldsymbol{G}_a^{\lambda|t} &= \boldsymbol{G}_a^{\lambda|t-1} + (\lambda\gamma)^{t-1-a}\boldsymbol{\delta}'_{a,t-1} \\
&= \left(\boldsymbol{G}_a^{\lambda|t-2} + (\lambda\gamma)^{t-2-a}\boldsymbol{\delta}'_{a,t-2}\right) + (\lambda\gamma)^{t-1-a}\boldsymbol{\delta}'_{a,t-1} \\
&= \boldsymbol{G}_a^{\lambda|a+1} + (\lambda\gamma)^1\boldsymbol{\delta}'_{a,a+1} + \cdots + (\lambda\gamma)^{t-1-a}\boldsymbol{\delta}'_{a,t-1} \\
&= \boldsymbol{G}_a^{\lambda|a+1} + \sum_{b=a+1}^{t-1}(\gamma\lambda)^{b-a}\boldsymbol{\delta}'_{a,b} && \square
\end{aligned}
$$

**Lemma A.3.** $\boldsymbol{G}_a^{\lambda|a+1} = \boldsymbol{\delta}'_{a,a} + g(\boldsymbol{h}_a;\boldsymbol{\theta}_{a-1})$

**Proof:**

$$
\begin{aligned}
\boldsymbol{G}_a^{\lambda|a+1} &= \boldsymbol{G}_a^1 \\
&= \frac{\partial L_{a+1}}{\partial \boldsymbol{h}_a} + \gamma g(\boldsymbol{h}_{a+1};\boldsymbol{\theta}_a)^\top\frac{\partial \boldsymbol{h}_{a+1}}{\partial \boldsymbol{h}_a} \\
&= \frac{\partial L_{a+1}}{\partial \boldsymbol{h}_a} + \gamma g(\boldsymbol{h}_{a+1};\boldsymbol{\theta}_a)^\top\frac{\partial \boldsymbol{h}_{a+1}}{\partial \boldsymbol{h}_a} - g(\boldsymbol{h}_t;\boldsymbol{\theta}_{t-1})^\top\frac{\partial \boldsymbol{h}_a}{\partial \boldsymbol{h}_a} + g(\boldsymbol{h}_a;\boldsymbol{\theta}_{a-1})^\top\frac{\partial \boldsymbol{h}_a}{\partial \boldsymbol{h}_a} \\
&= \boldsymbol{\delta}'_{a,a} + g(\boldsymbol{h}_a;\boldsymbol{\theta}_{a-1}) && \square
\end{aligned}
$$

**Lemma A.4.** $\sum_{b=a}^{t-1}(\gamma\lambda)^{b-a}\boldsymbol{\delta}'_{a,b} = \boldsymbol{G}_a^{\lambda|t} - g(\boldsymbol{h}_a;\boldsymbol{\theta}_{a-1})$

**Proof:**

$$
\begin{aligned}
\boldsymbol{G}_a^{\lambda|t} &= \boldsymbol{G}_a^{\lambda|a+1} + \sum_{b=a+1}^{t-1}(\gamma\lambda)^{b-a}\boldsymbol{\delta}'_{a,b} && \text{Lemma A.2} \\
&= \boldsymbol{\delta}'_{a,a} + g(\boldsymbol{h}_a;\boldsymbol{\theta}_{a-1}) + \sum_{b=a+1}^{t-1}(\gamma\lambda)^{b-a}\boldsymbol{\delta}'_{a,b} && \text{Lemma A.3} \\
&= \boldsymbol{\theta}_{a-1}^\top\boldsymbol{h}_a + \sum_{b=a}^{t-1}(\gamma\lambda)^{b-a}\boldsymbol{\delta}'_{a,b} && \square
\end{aligned}
$$

**Lemma A.5.** $e_b^{ji} = \sum_{a=0}^{b} (\gamma\lambda)^{b-a} \frac{\partial \boldsymbol{h}_b}{\partial \boldsymbol{h}_a} \frac{\partial g(\boldsymbol{h}_a; \boldsymbol{\theta}_a)}{\partial \theta_a^{ji}}$

**Proof:**

$$
\begin{aligned}
\boldsymbol{e}_b^{ji} &= \gamma\lambda \frac{\partial \boldsymbol{h}_b}{\partial \boldsymbol{h}_{b-1}} \boldsymbol{e}_{b-1}^{ji} + \frac{\partial g(\boldsymbol{h}_b; \boldsymbol{\theta}_b)}{\partial \theta_b^{ji}} \\
&= \gamma\lambda \frac{\partial \boldsymbol{h}_b}{\partial \boldsymbol{h}_{b-1}} \left( \gamma\lambda \frac{\partial \boldsymbol{h}_{b-1}}{\partial \boldsymbol{h}_{b-2}} \boldsymbol{e}_{b-2}^{ji} + \frac{\partial g(\boldsymbol{h}_{b-1}; \boldsymbol{\theta}_{b-1})}{\partial \theta_{b-1}^{ji}} \right) + \frac{\partial g(\boldsymbol{h}_b; \boldsymbol{\theta}_b)}{\partial \theta_b^{ji}} \\
&= (\gamma\lambda)^b \frac{\partial \boldsymbol{h}_b}{\partial \boldsymbol{h}_{b-1}} \frac{\partial \boldsymbol{h}_{b-1}}{\partial \boldsymbol{h}_{b-2}} \cdots \frac{\partial \boldsymbol{h}_1}{\partial \boldsymbol{h}_0} \frac{\partial g(\boldsymbol{h}_0; \boldsymbol{\theta})}{\partial \theta_0^{ji}} + (\gamma\lambda)^{b-1} \frac{\partial \boldsymbol{h}_b}{\partial \boldsymbol{h}_{b-1}} \cdots \frac{\partial \boldsymbol{h}_2}{\partial \boldsymbol{h}_1} \frac{\partial g(\boldsymbol{h}_1; \boldsymbol{\theta}_1)}{\partial \theta_1^{ji}} \\
&\quad + \cdots + \frac{\partial \boldsymbol{h}_b}{\partial \boldsymbol{h}_{b-1}} \frac{\partial g(\boldsymbol{h}_{b-1}; \boldsymbol{\theta}_{b-1})}{\partial \theta_{b-1}^{ji}} + \frac{\partial g(\boldsymbol{h}_b; \boldsymbol{\theta}_b)}{\partial \theta_b^{ji}} \\
&= (\gamma\lambda)^b \frac{\partial \boldsymbol{h}_b}{\partial \boldsymbol{h}_0} \frac{\partial g(\boldsymbol{h}_0; \boldsymbol{\theta}_0)}{\partial \theta_0^{ji}} + (\gamma\lambda)^{b-1} \frac{\partial \boldsymbol{h}_b}{\partial \boldsymbol{h}_1} \frac{\partial g(\boldsymbol{h}_1; \boldsymbol{\theta}_1)}{\partial \theta_1^{ji}} \\
&\quad + \cdots + (\gamma\lambda) \frac{\partial \boldsymbol{h}_b}{\partial \boldsymbol{h}_{b-1}} \frac{\partial g(\boldsymbol{h}_{b-1}; \boldsymbol{\theta}_{b-1})}{\partial \theta_{b-1}^{ji}} + \frac{\partial g(\boldsymbol{h}_b; \boldsymbol{\theta}_b)}{\partial \theta_b^{ji}} \\
&= \sum_{a=0}^{b} (\gamma\lambda)^{b-a} \frac{\partial \boldsymbol{h}_b}{\partial \boldsymbol{h}_a} \frac{\partial g(\boldsymbol{h}_a; \boldsymbol{\theta}_a)}{\partial \theta_a^{ji}} \qquad\qquad\qquad \square
\end{aligned}
$$

**Lemma A.6.** $\sum_{b=a}^{t-1} (\gamma\lambda)^{b-a} \boldsymbol{\delta}_{a,b} = \boldsymbol{G}_a^{\lambda|t} - g(\boldsymbol{h}_a; \boldsymbol{\theta}_{a-1}) + \mathcal{O}(\alpha)$

**Proof:** Using $\boldsymbol{\delta}_{a,t} = \boldsymbol{\delta}_{a,t}' - g(\boldsymbol{h}_t; \boldsymbol{\theta}_t)^\top \frac{\partial \boldsymbol{h}_t}{\partial \boldsymbol{h}_a} + g(\boldsymbol{h}_t; \boldsymbol{\theta}_{t-1})^\top \frac{\partial \boldsymbol{h}_t}{\partial \boldsymbol{h}_a}$, we have

$$
\begin{aligned}
\sum_{b=a}^{t-1} (\gamma\lambda)^{b-a} \boldsymbol{\delta}_{a,b} &= \sum_{b=a}^{t-1} (\gamma\lambda)^{b-a} \left( \boldsymbol{\delta}_{a,t}' - g(\boldsymbol{h}_t; \boldsymbol{\theta}_t)^\top \frac{\partial \boldsymbol{h}_t}{\partial \boldsymbol{h}_a} + g(\boldsymbol{h}_t; \boldsymbol{\theta}_{t-1})^\top \frac{\partial \boldsymbol{h}_t}{\partial \boldsymbol{h}_a} \right) \\
&= \sum_{b=a}^{t-1} (\gamma\lambda)^{b-a} \boldsymbol{\delta}_{a,b}' - \sum_{b=a}^{t-1} (\gamma\lambda)^{b-a} \left( g(\boldsymbol{h}_t; \boldsymbol{\theta}_t) - g(\boldsymbol{h}_{t-1}; \boldsymbol{\theta}_{t-1}) \right)^\top \frac{\partial \boldsymbol{h}_b}{\partial \boldsymbol{h}_a} \\
&= \sum_{b=a}^{t-1} (\gamma\lambda)^{b-a} \boldsymbol{\delta}_{a,b}' + \mathcal{O}(\alpha) \\
&= \boldsymbol{G}_a^{\lambda|t} - g(\boldsymbol{h}_a; \boldsymbol{\theta}_{a-1}) + \mathcal{O}(\alpha) \qquad\qquad \text{Lemma A.4} \quad \square
\end{aligned}
$$

**Proposition A.7.** *Let $\boldsymbol{\theta}_0$ be the initial weight vector and let $\theta^{ji}$ be an arbitrary parameter of $\boldsymbol{\theta}$. $\theta_t^{ji}$ be the weight at time $t$ computed by accumulate BP($\lambda$), and $\theta_t^{t,ji}$ be the weight at time $t$ computed by the online $\lambda$ -SG algorithm. Furthermore, assume that $\sum_{a=0}^{t-1} \Delta_a^{t,ji} \neq 0$, where $\Delta_a^{t,ji} := \left( \boldsymbol{G}_{a,0}^{\lambda|t} - g(\boldsymbol{h}_a; \boldsymbol{\theta}_0) \right)^\top \frac{\partial g(\boldsymbol{h}_a; \boldsymbol{\theta}_a)}{\partial \theta_a^{ji}}$ and $\boldsymbol{G}_{a,0}^{\lambda|t}$ is the $\lambda$-weighted synthetic gradient which uses $\boldsymbol{\theta}_0$ for all synthetic gradient estimations. Then, for all time steps $t$:*

$$
\frac{\left\| \theta_t^{ji} - \theta_t^{t,ji} \right\|_2}{\left\| \theta_t^{ji} - \theta_0^{ji} \right\|_2} \to 0 \quad as \quad \alpha \to 0
$$

.

**Proof:** The updates according to accumulate BP($\lambda$) (Equations 19 to 20) follow

$$\theta_t^{ji} = \theta_0^{ji} + \alpha \sum_{b=0}^{t-1} \boldsymbol{\delta}_b^\top \boldsymbol{e}_b \qquad\qquad \text{Definition: Equation 18}$$

$$= \theta_0^{ji} + \alpha \sum_{b=0}^{t-1} \boldsymbol{\delta}_b^\top \left( \sum_{a=0}^{b} (\gamma\lambda)^{b-a} \frac{\partial \boldsymbol{h}_b}{\partial \boldsymbol{h}_a} \frac{\partial g(\boldsymbol{h}_a; \boldsymbol{\theta}_a)}{\partial \theta_a^{ji}} \right) \qquad\qquad \text{Lemma A.5}$$

$$= \theta_0^{ji} + \alpha \sum_{b=0}^{t-1} \sum_{a=0}^{b} (\gamma\lambda)^{b-a} \boldsymbol{\delta}_b^\top \frac{\partial \boldsymbol{h}_b}{\partial \boldsymbol{h}_a} \frac{\partial g(\boldsymbol{h}_a; \boldsymbol{\theta}_a)}{\partial \theta_a^{ji}}$$

$$= \theta_0^{ji} + \alpha \sum_{b=0}^{t-1} \sum_{a=0}^{b} (\gamma\lambda)^{b-a} \boldsymbol{\delta}_{a,b}^\top \frac{\partial g(\boldsymbol{h}_a; \boldsymbol{\theta}_a)}{\partial \theta_a^{ji}} \qquad\qquad \text{Definition: Equation 21}$$

$$= \theta_0^{ji} + \alpha \sum_{a=0}^{t-1} \left( \sum_{b=a}^{t-1} (\gamma\lambda)^{b-a} \boldsymbol{\delta}_{a,b}^\top \right) \frac{\partial g(\boldsymbol{h}_a; \boldsymbol{\theta}_a)}{\partial \theta_a^{ji}} \qquad\qquad \sum_{b=k}^{n}\sum_{a=k}^{b} x_{a,b} = \sum_{a=k}^{n}\sum_{b=a}^{n} x_{a,b}$$

$$= \theta_0^{ji} + \alpha \sum_{a=0}^{t-1} \left( \boldsymbol{G}_a^{\lambda|t} - g(\boldsymbol{h}_a; \boldsymbol{\theta}_{a-1}) + \mathcal{O}(\alpha) \right)^\top \frac{\partial g(\boldsymbol{h}_a; \boldsymbol{\theta}_a)}{\partial \theta_a^{ji}} \qquad\qquad \text{Lemma A.6}$$

$$= \theta_0^{ji} + \alpha \sum_{a=0}^{t-1} \left( \boldsymbol{G}_{a,0}^{\lambda|t} - g(\boldsymbol{h}_a; \boldsymbol{\theta}_0) + \mathcal{O}(\alpha) \right)^\top \frac{\partial g(\boldsymbol{h}_a; \boldsymbol{\theta}_a)}{\partial \theta_a^{ji}}.$$

Where the last line holds as $\alpha \to 0$. On the other hand, the updates according to the online $\lambda$-SG algorithm produce

$$\theta_t^{t,ji} = \theta_0^{ji} + \alpha \sum_{a=0}^{t-1} \left( \boldsymbol{G}_a^{\lambda|t} - g(\boldsymbol{h}_a; \boldsymbol{\theta}_a) \right)^\top \frac{\partial g(\boldsymbol{h}_a; \boldsymbol{\theta}_a)}{\partial \theta_a^{ji}} \qquad\qquad \text{Definition: Equation 13}$$

$$= \theta_0^{ji} + \alpha \sum_{a=0}^{t-1} \left( \boldsymbol{G}_{a,0}^{\lambda|t} - g(\boldsymbol{h}_a; \boldsymbol{\theta}_0) + \mathcal{O}(\alpha) \right)^\top \frac{\partial g(\boldsymbol{h}_a; \boldsymbol{\theta}_a)}{\partial \theta_a^{ji}}.$$

Hence

$$\frac{\left\| \theta_t^{ji} - \theta_t^{t,ji} \right\|_2}{\left\| \theta_t^{ji} - \theta_0^{ji} \right\|_2} = \frac{\left\| \left( \theta_t^{ji} - \theta_t^{t,ji} \right)/\alpha \right\|_2}{\left\| \left( \theta_t^{ji} - \theta_0^{ji} \right)/\alpha \right\|_2} = \frac{\mathcal{O}(\alpha)}{C + \mathcal{O}(\alpha)}$$

where

$$C = \left\| \sum_{a=0}^{t-1} \left( \boldsymbol{G}_{a,0}^{\lambda|t} - g(\boldsymbol{h}_a; \boldsymbol{\theta}_0) \right)^\top \frac{\partial g(\boldsymbol{h}_a; \boldsymbol{\theta}_a)}{\partial \theta_a^{ji}} \right\|_2 = \| \sum_{a=0}^{t-1} \Delta_a^{t,ji} \|_2.$$

From the condition that $\sum_{a=0}^{t-1} \Delta_a^{t,ji} \neq 0$ we have $C > 0$. Therefore

$$\frac{\left\| \theta_t^{ji} - \theta_t^{t,ji} \right\|_2}{\left\| \theta_t^{ji} - \theta_0^{ji} \right\|_2} \to 0 \quad \text{as} \quad \alpha \to 0 \qquad \square$$

## B   Recursive definition of $G_t^\lambda$

In Proposition B.2 we provide a recursive definition of the $\lambda$-weighted synthetic gradient $\boldsymbol{G}_t^\lambda$ as defined in Equation 11. Note that this is similar, but not the same, as the recursive definition defined in A.3 in the

supplementary material of Jaderberg et al. (2017). In particular, this definition strictly considers future losses as in the context of a synthesiser target, whilst that provided in Jaderberg et al. (2017) also considers the loss at the current timestep $(L_t)$.

We first require the following Lemma

**Lemma B.1.** *For any sequence* $\{x_n\}_{n \geq 1}$,

$$\sum_{n=1}^{\infty} \lambda^{n-1} x_n = (1 - \lambda) \sum_{n=1}^{\infty} \lambda^{n-1} \sum_{k=1}^{n} x_k. \tag{23}$$

**Proof:** By the sum rule,

$$(1 - \lambda) \sum_{n=1}^{\infty} \sum_{k=1}^{n} \lambda^{n-1} x_k = (1 - \lambda) \sum_{k=1}^{\infty} \sum_{n=k}^{\infty} \lambda^{n-1} x_k$$

Now,

$$\sum_{n=k}^{\infty} \lambda^{n-1} = \sum_{n=1}^{\infty} \lambda^{n-1} - \sum_{n=1}^{k-1} \lambda^{n-1}$$

$$= \sum_{n=0}^{\infty} \lambda^{n} - \sum_{n=0}^{k-2} \lambda^{n}$$

$$= \frac{1}{1 - \lambda} - \frac{1 - \lambda^{k-1}}{1 - \lambda}$$

$$= \frac{\lambda^{k-1}}{1 - \lambda}.$$

So

$$(1 - \lambda) \sum_{n=1}^{\infty} \sum_{k=1}^{n} \lambda^{n-1} x_k = (1 - \lambda) \sum_{k=1}^{\infty} \left( \sum_{n=k}^{\infty} \lambda^{n-1} \right) x_k$$

$$= (1 - \lambda) \sum_{k=1}^{\infty} \frac{\lambda^{k-1}}{1 - \lambda} x_k$$

$$= \sum_{k=1}^{\infty} \lambda^{k-1} x_k$$

$$= \sum_{n=1}^{\infty} \lambda^{n-1} x_n \qquad \square$$

**Proposition B.2.** *We can define the* $\lambda$-*weighted synthetic gradient (Eq. 11) incrementally. In particular,*

$$\boldsymbol{G}_t^{\lambda} = \frac{\partial L_{t+1}}{\partial \boldsymbol{h}_t} + \gamma \lambda (\boldsymbol{G}_{t+1}^{\lambda})^{\top} \frac{\partial \boldsymbol{h}_{t+1}}{\partial \boldsymbol{h}_t} + \gamma (1 - \lambda) g(\boldsymbol{h}_{t+1}; \boldsymbol{\theta}_t)^{\top} \frac{\partial \boldsymbol{h}_{t+1}}{\partial \boldsymbol{h}_t}. \tag{24}$$

**Proof**: Let $T \leq \infty$ be the sequence length and define $\boldsymbol{G}_t^{(\tau)} := 0$ for $\tau > T$. Then we can write $\boldsymbol{G}_{t+1}^\lambda$ as

$$
\begin{aligned}
\boldsymbol{G}_{t+1}^\lambda &= (1-\lambda) \sum_{n=1}^{\infty} \lambda^{n-1} \boldsymbol{G}_{t+1}^{(n)} \\
&= (1-\lambda) \sum_{n=1}^{\infty} \lambda^{n-1} \left[ \sum_{k=1}^{n} \gamma^{k-1} \frac{\partial L_{t+1+k}}{\partial \boldsymbol{h}_{t+1}} + \gamma^n g(\boldsymbol{h}_{t+1+n}; \boldsymbol{\theta}_{t+n})^\top \frac{\partial \boldsymbol{h}_{t+1+n}}{\partial \boldsymbol{h}_{t+1}} \right] \\
&= (1-\lambda) \sum_{n=1}^{\infty} \lambda^{n-1} \sum_{k=1}^{n} \gamma^{k-1} \frac{\partial L_{t+1+k}}{\partial \boldsymbol{h}_{t+1}} + (1-\lambda) \sum_{n=1}^{\infty} \lambda^{n-1} \gamma^n g(\boldsymbol{h}_{t+1+n}; \boldsymbol{\theta}_{t+n})^\top \frac{\partial \boldsymbol{h}_{t+1+n}}{\partial \boldsymbol{h}_{t+1}} \\
&= \sum_{n=1}^{\infty} \lambda^{n-1} \gamma^{n-1} \frac{\partial L_{t+1+n}}{\partial \boldsymbol{h}_{t+1}} + (1-\lambda) \sum_{n=1}^{\infty} \lambda^{n-1} \gamma^n g(\boldsymbol{h}_{t+1+n}; \boldsymbol{\theta}_{t+n})^\top \frac{\partial \boldsymbol{h}_{t+1+n}}{\partial \boldsymbol{h}_{t+1}} \qquad \text{Lemma B.1} \\
&= \sum_{n=2}^{\infty} \lambda^{n-2} \gamma^{n-2} \frac{\partial L_{t+n}}{\partial \boldsymbol{h}_{t+1}} + (1-\lambda) \sum_{n=2}^{\infty} \lambda^{n-2} \gamma^{n-1} g(\boldsymbol{h}_{t+n}; \boldsymbol{\theta}_{t+n-1})^\top \frac{\partial \boldsymbol{h}_{t+n}}{\partial \boldsymbol{h}_{t+1}}.
\end{aligned}
$$

So

$$
\begin{aligned}
\boldsymbol{G}_t^\lambda &= (1-\lambda) \sum_{n=1}^{\infty} \lambda^{n-1} \boldsymbol{G}_t^{(n)} \\
&= (1-\lambda) \sum_{n=1}^{\infty} \lambda^{n-1} \left[ \sum_{k=1}^{n} \gamma^{k-1} \frac{\partial L_{t+k}}{\partial \boldsymbol{h}_t} + \gamma^n g(\boldsymbol{h}_{t+n}; \boldsymbol{\theta}_{t+n-1})^\top \frac{\partial \boldsymbol{h}_{t+n}}{\partial \boldsymbol{h}_t} \right] \\
&= (1-\lambda) \sum_{n=1}^{\infty} \lambda^{n-1} \sum_{k=1}^{n} \gamma^{k-1} \frac{\partial L_{t+k}}{\partial \boldsymbol{h}_t} + (1-\lambda) \sum_{n=1}^{\infty} \lambda^{n-1} \gamma^n g(\boldsymbol{h}_{t+n}; \boldsymbol{\theta}_{t+n-1})^\top \frac{\partial \boldsymbol{h}_{t+n}}{\partial \boldsymbol{h}_t} \\
&= \sum_{n=1}^{\infty} \lambda^{n-1} \gamma^{n-1} \frac{\partial L_{t+n}}{\partial \boldsymbol{h}_t} + (1-\lambda) \sum_{n=1}^{\infty} \lambda^{n-1} \gamma^n g(\boldsymbol{h}_{t+n}; \boldsymbol{\theta}_{t+n-1})^\top \frac{\partial \boldsymbol{h}_{t+n}}{\partial \boldsymbol{h}_t} \qquad \text{Lemma B.1} \\
&= \frac{\partial L_{t+1}}{\partial \boldsymbol{h}_t} + \lambda\gamma \left[ \sum_{n=2}^{\infty} \lambda^{n-2} \gamma^{n-2} \frac{\partial L_{t+n}}{\partial \boldsymbol{h}_{t+1}} \right] \frac{\partial \boldsymbol{h}_{t+1}}{\partial \boldsymbol{h}_t} + (1-\lambda)\gamma g(\boldsymbol{h}_t; \boldsymbol{\theta}_t)^\top \frac{\partial \boldsymbol{h}_{t+1}}{\partial \boldsymbol{h}_t} \\
&\quad + \lambda\gamma \left[ (1-\lambda) \sum_{n=2}^{\infty} \lambda^{n-2} \gamma^{n-1} g(\boldsymbol{h}_{t+n}; \boldsymbol{\theta}_{t+n-1})^\top \frac{\partial \boldsymbol{h}_{t+n}}{\partial \boldsymbol{h}_{t+1}} \right] \frac{\partial \boldsymbol{h}_{t+1}}{\partial \boldsymbol{h}_t} \\
&= \frac{\partial L_{t+1}}{\partial \boldsymbol{h}_t} + \lambda\gamma \left[ \sum_{n=2}^{\infty} \lambda^{n-2} \gamma^{n-2} \frac{\partial L_{t+n}}{\partial \boldsymbol{h}_{t+1}} + (1-\lambda) \sum_{n=2}^{\infty} \lambda^{n-2} \gamma^{n-1} g(\boldsymbol{h}_{t+n}; \boldsymbol{\theta}_{t+n-1})^\top \frac{\partial \boldsymbol{h}_{t+n}}{\partial \boldsymbol{h}_t} \right] \frac{\partial \boldsymbol{h}_{t+1}}{\partial \boldsymbol{h}_t} \\
&\quad + (1-\lambda)\gamma g(\boldsymbol{h}_{t+1}; \boldsymbol{\theta}_t)^\top \frac{\partial \boldsymbol{h}_{t+1}}{\partial \boldsymbol{h}_t} \\
&= \frac{\partial L_{t+1}}{\partial \boldsymbol{h}_t} + \gamma\lambda (\boldsymbol{G}_{t+1}^\lambda)^\top \frac{\partial \boldsymbol{h}_{t+1}}{\partial \boldsymbol{h}_t} + \gamma(1-\lambda) g(\boldsymbol{h}_{t+1}; \boldsymbol{\theta}_t)^\top \frac{\partial \boldsymbol{h}_{t+1}}{\partial \boldsymbol{h}_t} \qquad\qquad \square
\end{aligned}
$$

## C  Experimental Details

For the experiments which analyse the alignment of synthetic gradients to true gradients (Figures 2 and 6), we use an RNN with a linear activation function. For the experiments for which we train the RNN using synthetic gradients (Figures 4 and 5), we use LSTM units in our RNN architecture (Hochreiter & Schmidhuber, 1997).

The model output for a given task is a (trained) linear readout of the RNN (output) state. The synthesiser network performs a linear operation (with a bias term) on the RNN hidden state (concatenation of cell state and output state for LSTM) to produce an estimate of its respective loss gradient. The synthetic gradient at the final timestep is defined as zero. As in Jaderberg et al. (2017), the synthesiser parameters $\theta$ are initialised at zero.

When truncated BPTT with truncation size $n$ is used for a task sequence length $T$, the sequence is divided such that the first truncation is of size $R$ where $R = T \mod n$ and each following truncation is of size $n$.

For (accumulate) BP($\lambda$) we often find it empirically important to use a discount factor $\gamma < 1$. Except in the experiments using fixed RNNs (Fig. 2) for which $\gamma = 1$, for all conditions we set $\gamma = 0.9$.

We often observe it necessary to scale down the synthetic gradient before it is received by the RNN. We find this particularly necessary when using synthetic gradients alongside truncated BPTT, and in this case generally find a scaling factor 0.1 optimal as in Jaderberg et al. (2017). We find that BP($\lambda$) is less sensitive to this condition and choose scaling factors depending on the task (see below).

Data is provided to the model in batches. Though it is in principle possible to update the model as soon as the synthetic gradient is provided (e.g. every timestep), we find that this can lead to unstable learning, particularly when explicit supervised signals are sparse (as in the sequential-MNIST task). For this reason we instead accumulate gradients over timesteps and update the model at the end of the batch sequence. We use an ADAM optimiser for gradient descent on the model parameters (Kingma & Ba, 2014).

All experiments are run using the PyTorch library. Code used for the experiments can be found on the Github page: `https://github.com/neuralml/bp_lambda`.

The toy task experiments used to analyse gradient alignment were conducted with an Intel i7-8665U CPU, where each run with a particular seed took approximately one minute. The sequential MNIST task and copy-repeat tasks were conducted on NVIDIA GeForce RTX 2080 Ti GPUs. Each run in the sequential MNIST task took approximately 3 hours or less (depending on the model used); each run in the copy-repeat task took approximately 12 hours or less.

### C.1  Toy task used to analyse gradient alignment

To analyse gradient alignment for fixed RNNs in the toy task we provide the model an input at timestep 1 and the model is trained to produce a target 2-d coordinate on the unit-circle at the final timestep $T$ where $T = 10$. The input is a (fixed) randomly generated binary vector of dimension 10. The task error is defined as the mean-squared error between the model output and target coordinate.

We also consider plastic RNNs which are themselves updated at the same time as the synthesiser. To ensure that the RNN has to discover the temporal association between input and target, and not simply readout a fixed value, in this case we consider 3 input/target pairs, where the targets are spread equidistantly on the unit circle. For this case we are interested in the limits of temporal association that can be capture and consider increasing sequence lengths $T$; specifically, we consider $T$ as multiples of 10 up to 100. We deem a sequence length solved by a model with a certain initialisation (i.e. a given seed) if the model is able to achieve less that 0.025 – averaged over the last 20 of 250 training epochs – for that sequence length and all smaller sequence lengths. To improve readability, for this task the training curves are smoothed using a Savitzky–Golay filter (with a filter window of length 25 and polynomial of order 3).

For this task we provide the model in batches of size 10, where 1 epoch involves 100 batches. The number of RNN units is 30 and the initial learning rate for the synthesiser is set as $1 \times 10^{-4}$ and $1 \times 10^{-3}$ for the fixed and plastic RNN cases respectively.

Finally, to elucidate whether, like the TD($\lambda$) algorithm (see e.g. Figure 2 in Van Seijen et al. (2016)), BP($\lambda$) can only operate for modest learning rates, we test the model with different step sizes $\alpha$ and compare the resulting performance against the online $\lambda$-SG algorithm (Figure 3). Here we consider a plastic RNN trained using the respective synthetic gradients with the task sequence length $T = 10$. Note that in this case synthesiser weights are updated at each timestep (Eq. 14) as opposed to only at the end of the sequence once all gradients are accumulated. For these experiments standard stochastic gradient descent is used.

### C.2  Sequential-MNIST task

For the sequential-MNIST task we present a row-by-row presentation of a given MNIST image to the model (Deng, 2012; Le et al., 2015). That is, the task is divided into 28 timesteps where at the $i$th timestep the

$i$th row of 28 pixels is provided to the model. At the end of the presentation, the model must classify which digit the input represents. The task loss is defined as the cross entropy between the model output and the target digit.

For this task we provide the model in batches of size 50 with an initial learning rate of $3 \times 10^{-4}$. The number of hidden LSTM units is 30. For BP($\lambda$) the synthetic gradient is scaled by a factor of 0.1.

During training, the models with the lowest validation score over 50 epochs are selected to produce the final test error.

To verify that BP($\lambda$) can operate on non-trivial tasks with simple learning optimizers, we also perform weight updates for this task with stochastic gradient descent with momentum term $\zeta = 0.9$ (Fig. 9). In this case the learning rates for each network is fixed at 0.01.

### C.3 Copy-repeat task

For the copy-repeat task (Graves et al., 2014) the model receives a delimiter before an 8-dimensional binary pattern of length $N$ and then a repeat character $R$ The model must then repeat the binary pattern $R$ times followed by a stop character. The total sequence length is therefore $N \times (R + 1) + 3$. For easier absorption $R$ is normalised by 10 when consumed by the model.

We follow the curriculum in Jaderberg et al. (2017) and alternatively increment $N$ and $R$ when a batch average less than 0.15 bits is achieved.

For this task we provide the model in batches of size 100. For the RNN and readout parameters we use an initial learning rate of $1 \times 10^{-3}$, whilst we find a smaller learning rate of $1 \times 10^{-5}$ for the synthesiser parameters necessary for stable learning. The number of hidden LSTM units is 100. For BP($\lambda$) we do not apply scaling to the synthetic gradient.

## D    Supplementary Figures and Tables

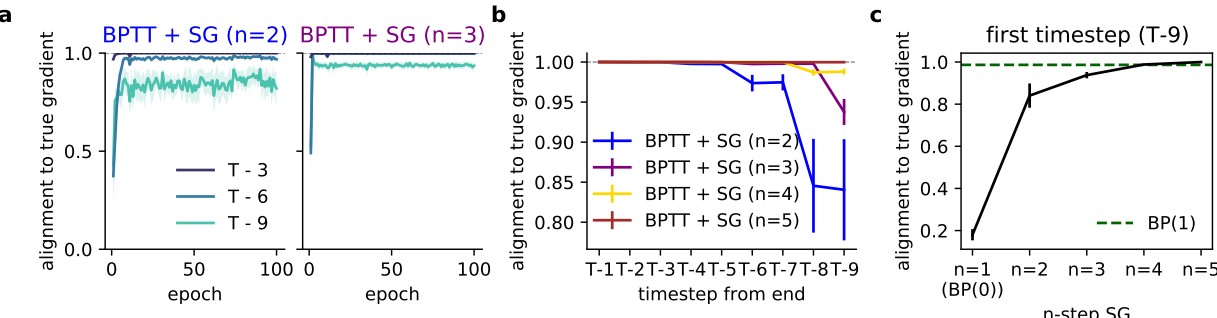

Figure 6: **Learning synthetic gradients with the $n$-step synthetic gradient (Eq. 9) as in (Jaderberg et al., 2017) in the toy task**. Inputs are provided at timestep 1 and the corresponding target is only available at the end of the task at time $T = 10$. (**a**) Alignment between synthetic gradients and true gradients for a fixed RNN model across different timesteps within the task, where the synthetic gradients are learned with BPTT truncation size $n = 2$ (left) and $n = 3$ (right). Alignment is defined using the cosine similarity metric. (**b**) The average alignment over the last 10% of epochs in **a** across all timesteps. (**c**) Alignment of synthetic gradients at the first timestep of the task for different $n$; BP(1) (dotted green) is shown for reference. Results show average (with SEM) over 5 different initial conditions.

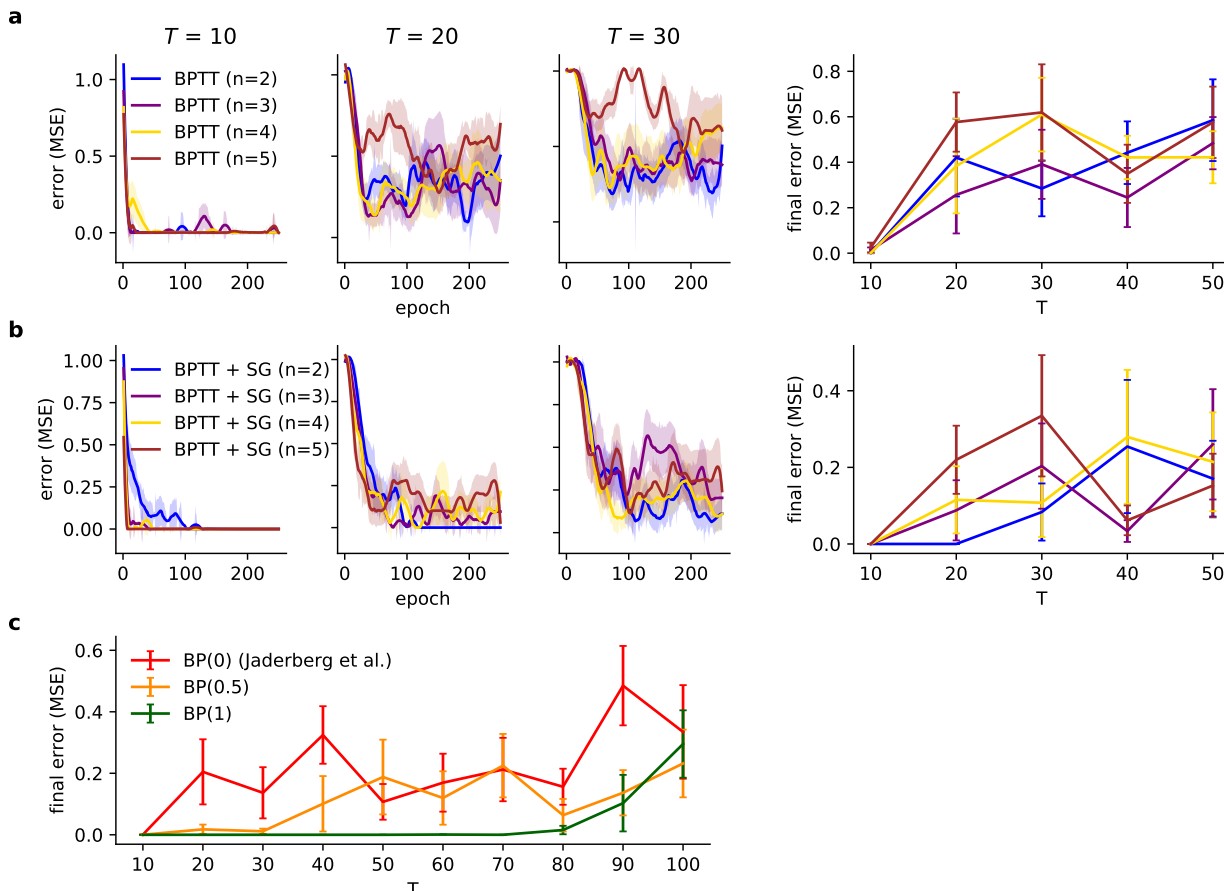

Figure 7: **RNN learning using $n$-step synthetic gradients in the toy task**. (**a**) (Left) Learning curves of RNNs which are updated using $n$-step truncated BPTT over different sequence lengths $T$. (Right) Task error at end of training for increasing $T$. (**b**) Same as (a) but $n$-step synthetic gradients are also applied. (**c**) Task error for RNNs updated using BP($\lambda$) at end of training for increasing $T$. Results show average (with SEM) over 5 different initial conditions.

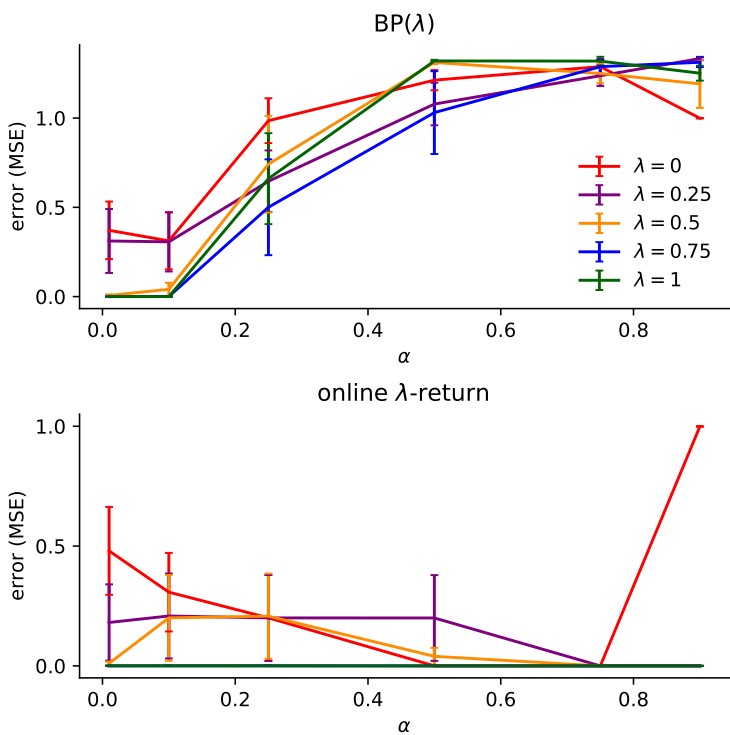

Figure 8: **Effect of step-size $\alpha$ on performance in toy task for (top) BP($\lambda$) and (bottom) the online $\lambda$-SG algorithm**. Task errors are averaged over the last 10 epochs of training (out of 100). Results show average (with SEM) over 5 different initial conditions.

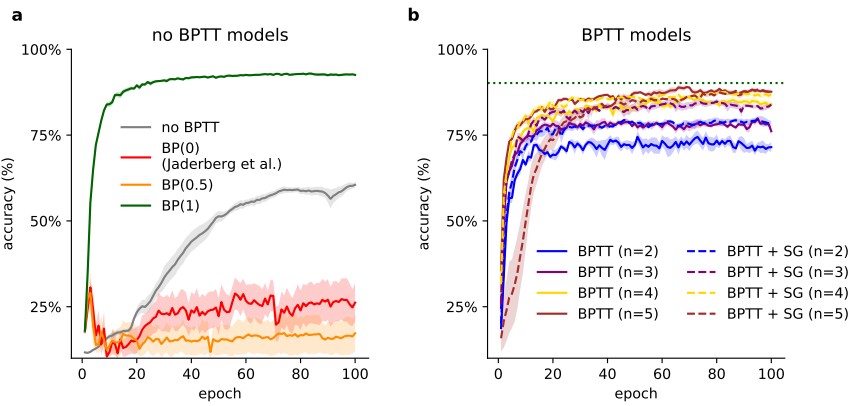

Figure 9: **BP($\lambda$) in sequential MNIST task with stochastic gradient descent**. (**a**) Validation accuracy during training for BP($\lambda$) models. (**b**) Validation accuracy during training for models which learn synthetic gradients (SG) with $n$-step truncated BPTT as in original implementation (Jaderberg et al., 2017); final performance of BP(1) (as in (b); dotted green) is given for reference. Results show average (with SEM) over 5 different initial conditions.

