# OpenReview forum: "BP($\mathbf{\lambda}$): Online Learning via Synthetic Gradients"
_TMLR — Accepted by TMLR_

### Review · Reviewer_3HWT · 2024-02-01

**Summary Of Contributions:**

The paper investigates the idea of discounted synthetic gradients as an alternative to BPTT when training a recurrent model, which is summarized in Figure 1 and section 2.1/2.2/2.3. The idea of discounted synthetic gradients is mentioned in the supplementary material of Jaderberg et al. (2017) but not experimentally evaluated. The paper under submission proposes an algorithm for learning these discounted gradients and experimentally validates them for sequential MNIST and a copy-repeat task.

**Audience:**

Yes

**Claims And Evidence:**

Yes

**Requested Changes:**

I am very open to discussing the weakness I raise on the experimental results not being directly comparable to previously published results

**Strengths And Weaknesses:**

# Strengths
+ The supplementary material in Jaderberg et al. (2017) mentioned this idea as it is reasonable and can help incorporate longer amounts of temporal information into the gradient estimates without needing to backprop through time for all of them.
+ Table 1 comparing the concepts between RL and synthetic gradients is a nice analogy to have (minor comment: instead of describing the rows in text it also seems like there is space to describe them inline and may slightly improve the readability)
+ Overall the paper is clear, well-written, and well-motivated.
+ The toy experimental results in Fig 2/3 are convincing and clearly validates the claim of being able to better approximate the true gradient.

# Weaknesses
+ The only weakness I see is that the experimental results on MNIST/copy-repeat are not directly comparable to results in Jaderberg et al. (2017), and also does not include all of their experimental settings (e.g., Penn Treebank). For this reason, I borderline mark the paper as not validating the claim as I do not easily see how to compare the experimental results with previously published results on these datasets

---

> ### Author Response · Authors · 2024-03-05
>
> “The only weakness I see is that the experimental results on MNIST/copy-repeat are not directly comparable to results in Jaderberg et al. (2017), and also does not include all of their experimental settings (e.g., Penn Treebank). For this reason, I borderline mark the paper as not validating the claim as I do not easily see how to compare the experimental results with previously published results on these datasets”
>
> The reviewer is correct that the presented experimental results are not directly comparable to those in Jadeberg et al. 2017, since (i) our implementation of the original synthetic gradient algorithm performs worse in the copy-repeat task (i.e. our models on average solve sequence lengths around half of the originally presented results) and (ii) we do not include results for the Penn Treebank task.
>
> With regards to (i) we believe this difference comes due to the disparity in hyperparameter choices. Most notably, Jadeberg et al. employed much larger RNN networks (256 vs 100 units); though not explicitly stated, given their other experiments it is also likely their synthesiser network was of a larger size (e.g. with hidden layers) than our implementation which is simply a linear function. The reason for our more modest hyperparameter choices is primarily due to the greater computational expense of our algorithm. As stated in the Limitations section, even with a linear synthesiser, the memory and computational complexity of BP$(\lambda)$ is $O(|h|^3)$ and $O(|h|^4)$ respectively, which for even modest network sizes is significantly greater than the memory/complexity of the original algorithm at $O(|h|^2)$ and $O(|h|^2)$ respectively (this is not made clear in the new Table 3). Given that we were primarily interested in demonstrating the benefit of more accurate synthetic gradients in these experiments, rather than achieving state of the art results, we therefore felt that a more practical choice of hyperparameters was justified. Finally, we also point out that our implementation is more consistent with those obtained in Bellec et al. 2019 (see their Fig 4e). We now note this in the manuscript (end of Section 4.3).
>
> With regards to (ii), we clarify to the reviewer that we have indeed attempted to use BP($\lambda$) on the Penn Treebank (both character prediction as per Jadeberg et al. 2017 and also word prediction. However, we have not been able to obtain positive results, with BP($\lambda$) often outperformed even by standard truncated BPTT. Like the copy-repeat task, we believe higher performance might be obtained if we significantly increase the number of parameters in our model; for example, in Jaderberg et al. 1024 units are used in the RNN. But again, the computational expense of our algorithm makes this somewhat impractical to explore.
>
> We hope that this inability to replicate and surpass the original experimental results does not detract from the strong empirical improvements we do see with BP(\lambda) over the original implementation for synthetic gradients, nor indeed the overall theoretical progress made in this work. We believe that our current results and theoretical arguments open a new window with great potential, which pushes the boundary of what is currently possible with synthetic gradients.

---

> > ### Comment · Reviewer_3HWT · 2024-03-05
> >
> > Thank you for the clarification on the experimental results there. It seems like a reasonable attempt at reproducing the results from Jaderberg et al. 2017, and is helpful to connect with Bellec et al., 2019. I've updated my evaluation of this paper to "meets all claims"

---

### Review · Reviewer_upTs · 2024-02-02

**Summary Of Contributions:**

The paper presents an alternative to back-propagation through time using synthetic gradients in an online manner.
The idea stems from the analogy between the original synthetic gradients approach and temporal difference algorithms developed in reinforcement learning. A particular feat of the method is to propose an online method that can update parameters of the synthetizer without the help of an intermediate backpropagation through time and that can provide more accurate estimates than the previously proposed approach. Several experiments illustrate the potential of the approach on synthetic tasks.

**Audience:**

Yes

**Claims And Evidence:**

Yes

**Requested Changes:**

Comments/questions/suggestions:
- I don't fully understand the last term in Eq 11. A priori it should be $G_t^{(T-t)}$ and it is not clear form the definition of $G_t^{(n)}$ that this matches $G_t$. It seems that it is rather $\partial L_T /\partial h_t$. I have probably missed something. But this means that some explanations would be beneficial.
- Line (13) is quite hard to follow: on one hand t would be the time step. On the other hand there is a index k, which, I suppose, is the iteration of the learner of $\theta$. In particular I would have thought that we would have $G_t^{\lambda|t}$ rather than $G_k^{\lambda|t}$. Similarly for the indexes of $h_k$. Again I may have missed something but explanations would be highly beneficial on that part.
- On that thread, it would be good to use an index k in line (7) instead of t which was use for time. (A priori the iterations fo the algorithms and the time steps of the RNN are two different things).
- The intuition on the proposed algorithm 3.4 is missing. Or, to be exact, it relies on knowledge from RL. The paper would highly benefit from a stand-alone explanation of the motivation of the algorithm.
- Having a table summarizing the computational and memory complexities of (i) the proposed approach, (ii) classical BPTT, (iii) BPTT with synthetic gradients previously proposed, would be beneficial for the paper.
- In the learning part of the experiments, what algorithm was used? SGD? Adam? How the method compares when using different optimizers?
- (Optional) In the experiments, it could be interesting to showcase the benefits in terms of memory of the proposed approach compared to BPTT. For example, if the architecture is such that the batch-size is ceiled to some number by BPTT, it could be interesting to see how the proposed algorithm can benefit from the relaxed memory requirements by using larger batch-sizes.

Minor comments:
- The legend in Table 1 is hard to follow. The authors may reorganize it in a more readable way.
- End of page 2: typo "...both empirically can it alleviate..."
- End of first paragraph of section 4.1: typo "...to learn perfectly model the true BPTT..."

**Strengths And Weaknesses:**

Strengths:
- The connection with reinforcement learning can bear numerous fruits. The proposed method is a clear example. Overall such an approach is both original and can inspire follow-ups.
- The idea is rather well explained at the start, though the final proposal is still hard to follow.
- The experiments illustrate clearly the potential of the approach.
- Limitations are clearly delineated.

Weaknesses:
- Unfortunately the final proposal is hard to follow. Some parts appear to lack some explanations (see suggestions below).
- A few more experiments could complete the exposition to give a full perspective on the idea (see suggestions below).
- The relevance of the approach beyond RNNs is not clear yet.

---

> ### Author Response · Authors · 2024-03-05
>
> Due to character limit restrictions we organise our answers to each of the points into two parts.
>
> **Part 1/2**:
>
> 1. “Unfortunately the final proposal is hard to follow. Some parts appear to lack some explanations (see suggestions below).”
>
> (i): “I don't fully understand the last term in Eq 11..”
> (ii): “Line (13) is quite hard to follow..”
> (iii): “On that thread, it would be good to use an index k..”
> (iv): “The intuition on the proposed algorithm 3.4 is missing..”
> (v): “Having a table summarizing the computational and memory complexities..”
>
> (i): The reviewer is correct that the last term is $G_t^{T-t}$. We can then apply eq (9) to see that $G_t^{T-t} = G_t$. Specifically, we have that $G_t^{T-t} = \sum_{t<\tau\leq T} \gamma^{\tau -t - 1} \frac{\partial L_{\tau}}{\partial h_t} + \gamma^{T-t} \hat{G}_T \frac{\partial h_T}{\partial h_t}$.
>
> Now, since the synthesiser at the final timestep is defined as zero, the second term here is zero and we have that $G_t^{T-t} = \sum_{t<\tau\leq T} \gamma^{\tau -t - 1} \frac{\partial L_{\tau}}{\partial h_t} = G_t$ . We hope that this has clarified things.
>
> (ii): The reviewer is correct that $k$ can be interpreted as an iterator for the updates the synthesiser weights $\theta$. In particular, at timestep $t$ of the sequence, $\theta$ undergoes as many updates by applying eq (13) for $k$ with $1 \leq k \leq n$. That is, at a given $t$ $\theta$ is applying an update based on each of the previously hidden RNN states and corresponding predictions. We confirm that the target value at iteration $k$ is based on $G_k^{\lambda|t}$ (incorporating all observed error gradients from k to t), not $G_t^{\lambda|t}$. Note that these iterative weight updates (and notation) is analogous to the online lambda-return algorithm in RL (see e.g. equations 6,7 in Seijen & Sutton, 2014). We have added text around eq (13) to clarify this.
>
> (iii): Following the above, we clarify to the reviewer that the iterator $k$ and timestep $t$ have important differences. t denotes the timestep at which the RNN has currently reached (i.e. error gradients beyond $t$ are unavailable), whilst k denotes the RNN state for which the synthesiser weights are updating. Since eq (9) in itself is agnostic about the time in the sequence and simply samples RNN states, the reviewer is correct that k could be used. However, we feel that $t$ here already suggests to the reader that this sampling could be time-dependent and based on RNN dynamics, as indeed is incorporated in our proposed algorithms. Moreover, like the above, we believed greater context could be inferred from our work by using the same standard notation as in RL (see Equation 1 in Seijen & Sutton, 2014). We have added text around eq (13) to clarify this.
>
> (iv): Thank you for the suggestion. We have now added a paragraph in section 3.4 which aims to provide more intuition to the algorithm.
>
> (v): Thank you for the suggestion. We have now added such a Table (Table 3) to the manuscript.

---

> ### Author Response · Authors · 2024-03-05
>
> **Part 2/2**
>
> 2. “A few more experiments could complete the exposition to give a full perspective on the idea (see suggestions below).”
>
> (i): “In the learning part of the experiments, what algorithm was used..”
>
> Following the implementation of synthetic gradients in Jadeberg et al. 2017, in all of the originally presented experiments we employed the Adam optimizer. However, we agree it is important to also demonstrate that the algorithm can work with other optimizers. To this end, we have now added new results for the sequential MNIST task using SGD (Fig 9). Moreover, we also explicitly examine the effect of the learning rate with SGD on the toy task in order to reveal interesting limitations of BP($\lambda$) for high learning rates (Fig 3c and Fig 8; see also new paragraph in Section 4.1).
>
> (ii) “(Optional) In the experiments, it could be interesting to showcase the benefits in terms of memory of the proposed approach compared to BPTT..”
>
> We agree with the reviewer that it would be interesting and indeed desirable to showcase the potential benefits of our algorithm compared to BPTT. However, the issue we face is that for a non-trivial amount of hidden units in the RNN the computational and memory complexity is simply too much greater than BPTT over reasonable periods of time or truncation sizes. For example, even for a modest RNN size of 30 units one would not see a memory advantage until a BPTT window of n > 30^2 is used. Unfortunately, tasks requiring such long sequences is currently out of the scope of this work.
>
> 3. “The relevance of the approach beyond RNNs is not clear yet.”
>
> The reviewer raises the important point that BP($\lambda$) is only presented within the framework of RNNs. We highlight that a key requisite of the algorithm is that the states of the network performing the task are sequentially dependent, since we require the Jacobian $dh_[t}/dh_{t-1}$ to be non-zero for eligibility traces to carry information over time (Eq 16). RNNs intrinsically satisfy this sequential dependency. We do note, however, that it is in principle possible to apply the same reasoning to feedforward networks, where we now interpret  a sequence of layers instead of sequence of timesteps; indeed, synthetic gradients were successfully applied in this way in Jaderberg et al. 2017. Note that in this case the task error would be defined as zero for all layers (timesteps) except for the last layer (timestep). However, since BP($\lambda$) operates by changing a universal weight vector for the synthesiser, one must assume that each hidden layer is of the same size and that the same synthesiser weights are used across layers. We have now added a paragraph discussing this in Section 6.

---

### Review · Reviewer_ReJk · 2024-02-15

**Summary Of Contributions:**

The paper builds upon the synthetic gradients framework and presents a new algorithm for learning synthetic gradients based on the idea of eligibility traces from RL. The new algorithm called accumulate BP($\lambda$) doesn't rely on BPTT and allows to control the bias-cost tradeoff and experimentally attains better results in terms of these two objectives.

**Audience:**

Yes

**Broader Impact Concerns:**

No concerns.

**Claims And Evidence:**

Yes

**Requested Changes:**

### Comparison with RTRL

Given that the computational requirement of Accumulate BP is similar to RTRL (Williams & Zipser, 1989), wouldn't it make sense to provide both theoretical and experimental comparison to this method?

### Variance analysis

As I understand, one reason why TD($\lambda$) is efficient is due to the ability of controlling bias-variance tradeoff in the presence of high-variance gradients. What is the situation with supervised learning? In your experiments, $\lambda=1$ seems to perform the best but it in the RL case that's usually not the case. If all we need is $\lambda=1$ then why bother with the RL origins of the method? I feel like I'm missing something here but it probably also suggests that this needs to discussed, especially since the paper draws parallels to RL.

Relatedly, does it make sense to have a plot similar to first one in Figure 2 of Seijen & Sutton, 2014? That would visualize not only dependency on $\lambda$ but also the learning rate. From that plot it is particularly clear that the optimal choice of $\lambda$ is tied to the choice of the learning rate.

### Notation and figures refinement
See my comments in the previous section

**Strengths And Weaknesses:**

## Strengths

I find the proposed algorithm very useful and interesting as any improvement on the ability of recurrent models to train online and approach performance of BPTT is important. The experiments, while arguably simplistic by the modern standards, follow the methodology of Jaderberg et al and do show a viable improvement due to the use of BP($\lambda$).

## Weaknesses

### Somewhat confusing notation

The paper could simply the notation and potentially eliminate some of the symbols ($v_t$ for example which doesn't seem to have its own significance). If it stays it needs a formal introduction in eqs 5 and 6 which should probably be coupled with $h_t$, otherwise if we assume $v_t$ comes from its own marginal distribution, I'm not sure the loss still makes sense. Sometimes the paper uses the partial derivative notation, sometimes nabla operator. The main text uses normal fonts for (could be-) vectors but these are written in bold in the supplementary material. All the different sub- and superscripts for $G$ require memorization while reading. This may confuse the reader.

Diagrams on Picture 1 should ideally be all vectorized.

I would also suggest to write all weight updates (such as eq 13) in the form of $\delta \theta_k^t = \ldots$, this will unify the style between the in-text equations and Algorithm 1 and possibly also improve readability.

### Some clarity issues

I find this statement contradictory: "for an appropriately small learningrate α the synthesiser learns the true BPTT gradient $G_t$" below Theorem 3.1. As I understand, it won't because $h_t$ likely doesn't have enough information to predict the true BPTT gradient.

### Plastic RNNs lack explanation

It would help if the exact plastic RNN model is referenced and ideally outlined at least briefly,  even if in the supplementary material.
If it's the model by Miconi et al, then there really needs to be a discussion of how synthetic gradients fit into this construction.

---

> ### Author Response · Authors · 2024-03-05
>
> Due to character limit restrictions we organise our answers to each of the points into two parts.
>
> **Part 1/2**
>
> 1. “The paper could simply the notation..”
>
> (i): “for example, v_t which doesn't seem to have its own significance..which should probably be coupled with h_t”
>
> Whilst we do believe it is important to define the notion of a target synthetic gradient, we agree the use of the new symbol $v_t$ is perhaps convoluted. We have now modified this to $\bar{G}_t$. Note that at this point we are agnostic about what conditions $\bar{G}_t$ should satisfy as we want to keep the notion of the target gradient as broad as possible. In particular, whilst $\bar{G}_t$ is the target synthetic gradient for the RNN state $h_t$, that does imply it must itself depend on $h_t$. For example, in the extreme case we want the synthesiser to have minimal influence, we could set  $\bar{G}_t = 0$ for all states $h_t$.
>
> (ii): “Sometimes the paper uses the partial derivative notation, sometimes nabla operator.”
>
> We agree this notation may be confusing to the reader. We have now modified the paper to use partial derivative notation throughout.
>
> (iii): “The main text uses normal fonts for (could be-) vectors but these are written in bold in the supplementary material”
>
> We decided to use normal fonts throughout the main text as we felt it improved the readability of the text and equations. However, if the reviewer feels strongly that this impacts interpretability then of course this can be easily changed. Indeed, we appreciate that multiple fonts can help remind the reader of the mathematical objects at play, which is why we distinguish vectors/tensors with bold font in the more technical results in the Appendix.
>
> 2. “Diagrams on Picture 1 should ideally be all vectorized”
>
> We prefer to use normal (non-bold) font to be consistent with the main text (see above comment). Again, we are happy to change this if the reviewer disagrees.
>
> 3. “I would also suggest to write all weight updates (such as eq 13) in the form of XX..”
>
> We believe that the slightly different form of weight updates in eq 13 (which now also depends on the iterator k, unlike e.g. eq 14) warrants this more explicit format which directly shows the dependence of one weight update to the next. Our worry is that by using the more compact delta notation this dependency and order of weight updates becomes less clear.
>
> 4. “I find this statement contradictory…the synthesiser learns the true BPTT gradient..it won't because (it) likely doesn't have enough information to predict the true BPTT gradient.”
>
> We appreciate the wording here is clumsy. Our point is that with a small enough learning rate \alpha BP($\lambda$) is equivalent to the synthesiser being trained with the true error gradient $G_t$ as the target value; that is, we have $\bar{G}_t = G_t$. Now, the reviewer is correct that this does not imply the synthesiser can successfully learn to exactly output this value, since the synthesiser is limited by the information in its input $h_t$ (e.g. the error gradient might depend on future task inputs independent of $h_t$). We have now reworded this to “the synthesiser learns with true BPTT gradient as its target, $\bar{G}_t =G_t$”.
>
> 5. “It would help if the exact plastic RNN model is referenced and ideally outlined at least briefly..”
>
> We are unsure what the reviewer is asking for here. The RNN architecture we use is a ‘vanilla’ RNN for the experiments on the toy dataset, and an LSTM network for the sequential-MNIST and copy-repeat tasks. The plastic RNN model in the toy experiments refers to the case where the RNN weights are updated (as opposed to fixed), where the updates are defined by the learning rule of the model (e.g. eq 4 for synthetic gradient models).

---

> > ### Author Response · Authors · 2024-03-05
> >
> > **Part 2/2**
> >
> > Requested Changes
> >
> > 1. “Comparison with RTRL..”
> >
> > For stability reasons, in all of our experiments the RNN weight gradients are accumulated over the sequence and the RNN update only occurs at the end. In this case RTRL is equivalent to full BPTT. We have performed full BPTT (and therefore RTRL) on the non-trivial tasks: for example the test accuracy for sequential MNIST is 97.5% and the average longest task sequence solved in the copy-repeat task is 65.8. We did not originally present these results as we wanted to highlight our comparison with other BPTT-constricted methods and synthetic gradient methods specifically. However, we would be happy to include these results under RTRL if the reviewer felt they would be useful.
> >
> > 2. “Variance analysis..”
> >
> > (i): “..What is the situation (of bias-variance tradeoff) with supervised learning?..”
> >
> > The reviewer is correct that a core aspect of TD(lambda) is the ability to control the bias-variance tradeoff; specifically, whilst a high lambda might reduce the bias in its return estimates, these estimates may be more prone to higher variance. We have previously examined whether such a situation might arise within our framework in supervised learning. However, one difficulty is that the behavior or indeed standard definition of bias and variance - each of which we currently assume to be represented by the cosine similarity metric - for predictions of vectors (not scalars) as for error gradients is more nuanced. For example, whilst bias (e.g. overestimation in return) in the prediction for one state in RL will directly lead to the same bias (overestimation) for the previous state, the hidden state jacobian in our framework (equation 15) means that how synthesiser bias, for example at the last RNN state, translates backwards through the previous states is significantly more complex/unpredictable. Indeed, we did not obtain meaningful empirical results when, for instance, we analysed the scalar bias for a given element of the true and predicted error gradient vectors. We also tried to directly analyse the intrinsic variance of the synthesiser predictions (this was taken as the mean variance of each element of the predicted vector); for example we predicted that a high lambda might suffer when learning noisy gradients in a noisy task). However, as with the presented results in the paper we almost always observed higher lambda to converge to the true gradient more quickly and therefore more stable/less varied predictions (i.e. the opposite of the predicted case in TD(lambda)). We have added text at the end of the first paragraph in Section 6 discussing this and the following point.
> >
> > (ii): “If all we need is \lambda=1 then why bother with the RL origins of the method?”
> >
> > The reviewer is right that in all of our originally presented experiments $\lambda=1$ achieves the best performance. We emphasise however that this case still requires (even more so to other cases) eligibility traces and therefore the theory of BP(\lambda) is necessary to formulate it. We do however acknowledge that it is perhaps surprising that \lambda=1 consistently achieves the best performance in our experiments, since in RL intermediate values may be preferable. We are unsure why this is the case, but speculate that there can often be more degrees of freedom in RL environments in terms of possible (environmental) states and task length, and thus perhaps some degree of bias (as implied with lambda < 1) is beneficial for convergence. We have added text at the end of the first paragraph in Section 6 discussing this point.
> >
> > (iii): “Relatedly, does it make sense to have a plot similar to first one in Figure 2 of Seijen & Sutton, 2014?..”
> >
> > We thank the reader for this good suggestion, and have now included an analogous plot to Figure 2 of Seijen & Sutton, 2014 (Figure 3c; see also added final paragraph in Section 4.1). On top of this, and related to the above comment, we have analysed the effect of the step-size across different values of \lambda (Figure 8); here for larger step-sizes we do see a slight but significant advantage of lower values for lambda compared to $\lambda = 1$, consistent with Figure 12.6 in Sutton and Barto, 1999 (second edition).
> >
> > Minor comments:
> >
> > 1. "The legend in Table 1 is hard to follow. The authors may reorganize it in a more readable way."
> >
> > We agree the original legend was perhaps overly cluttered and difficult to follow. We have now reduced its length and referred the reader to the main text and Seijen & Sutton, 2014 for a more detailed description of the different mathematical terms at play.
> >
> > 2. "End of page 2..End of first paragraph of section 4.1..(typos)"
> >
> > Fixed, thank you.

---

### Author Response · Authors · 2024-03-05
**Rebuttal**

Dear reviewers,

Thank you for the constructive comments and suggestions. We address each point raised individually below. We have also accordingly made changes to the manuscript, including changes to the text (in bold font) as well as the addition of one new main figure panel (Figure 3c) and two Supplementary figures (Figures 8,9).

---

### Decision · Action_Editor_GBaA · 2024-04-03

**Recommendation:** Accept as is

**Comment:**

The paper meets both criteria for acceptance. Also, it has already undergone a revision to incorporate reviewer feedback. The reviewers didn't request further changes after the rebuttal, so I don't see any reason for the paper to be revised again.

**Audience:**

The paper brings ideas from reinforcement learning into the synthetic-gradient framework for training RNNs. It takes an idea that was mentioned (but not explored) in previous work, and develops it both algorithmically and experimentally. This clearly meets TMLR "audience" criterion.

**Claims And Evidence:**

All three reviewers agree that the claims in the paper are supported by evidence. Reviewer 3HWT initially raised a concern regarding comparability of results with previous work, but this concern was allayed after the rebuttal.